# On the Second-order Convergence Properties of Random Search Methods

Aurelien Lucchi[*]    Antonio Orvieto    Adamos Solomou

Department of Computer Science
ETH Zurich

## Abstract

We study the theoretical convergence properties of random-search methods when optimizing non-convex objective functions without having access to derivatives. We prove that standard random-search methods that do not rely on second-order information converge to a second-order stationary point. However, they suffer from an exponential complexity in terms of the input dimension of the problem. In order to address this issue, we propose a novel variant of random search that exploits negative curvature by only relying on function evaluations. We prove that this approach converges to a second-order stationary point at a much faster rate than vanilla methods: namely, the complexity in terms of the number of function evaluations is only linear in the problem dimension. We test our algorithm empirically and find good agreements with our theoretical results.

## 1 Introduction

We consider solving the non-convex optimization problem $\min_{\mathbf{x} \in \mathbb{R}^d} f(\mathbf{x})$, where $f(\cdot)$ is differentiable but its derivatives are not directly accessible, or can only be approximated at a high computational cost. This setting recently gained attention in machine learning, in areas such as black-box adversarial attacks [11], reinforcement learning [51], meta-learning [7], online learning [9], and conditional optimization [57].

We focus our attention on a popular class of derivative-free methods known as random direct-search methods of directional type [1]. These methods optimize $f(\cdot)$ by evaluating the objective function over a number of (fixed or randomized) directions, to ensure descent using a sufficiently small stepsize. Direct-search algorithms date to the 1960's, including e.g. [44, 46]. More recent variants include deterministic direct search [15], random direct search (e.g. [54], or the Stochastic Three Points (STP) method [6]), which randomly sample a direction and accept a step in this direction if it decreases the function $f(\cdot)$. As discussed in [40], direct-search methods have remained popular over the years for a number of reasons, including their good performance and known global convergence guarantees [54], as well as their straightforward implementation that makes them suitable for many problems. We refer the reader to [40, 15] for a survey.

In machine learning, objective functions of interest are often non-convex, which poses additional challenges due to the presence of saddle points and potentially suboptimal local minima [33]. Instead of aiming for a global minimizer, one often seeks a second-order stationary point (SOSP): i.e. a solution where the gradient vanishes and the Hessian is positive definite. Indeed, as shown by [14, 36, 22, 21], many machine learning problems have no spurious local minimizers, yet have many saddle points which yield suboptimal solutions and are often hard to escape from [19]. While convergence to SOSPs and saddle escape times have been extensively studied in the context of

---

[*]Alphabetical ordering, all authors contributed equally.

[1]In this manuscript, we will use the terms "random direct-search" and "random search" interchangeably.

35th Conference on Neural Information Processing Systems (NeurIPS 2021).

gradient-based methods [32, 16, 10, 56], *prior analyses for (random) direct search have mainly focused on convergence to first-order stationary points* [6, 54]. One exception is the Approximate Hessian Direct Search (AHDS) method [26], that explicitly computes the Hessian of the objective to obtain second-order worst-case guarantees. However, computing or storing a full Hessian is prohibitively expensive in high dimensions.

Towards a better understanding of the complexity of finding second-order stationary points with random search methods, we make the following contributions:

- We study the complexity of a simple random search similar to STP (Algorithm 1) to reach SOSPs. We find that the (worst-case) complexity requires a number of function evaluations that scales *exponentially* in terms of the problem dimension $d$. As we will see, the exponential scaling is not an artefact of the analysis. This is intuitive, indeed, if we are at a saddle point where we just have one direction of negative curvature, finding good alignment with a *random* direction becomes exponentially difficult as the dimension of the space increases.

- To solve this issue, we design a variant of random search (RSPI, Algorithm 2) that, instead of randomly sampling directions from the sphere, relies on an approximate derivative-free power iteration routine to extract negative curvature direction candidates (unlike [26] that requires an approximation to the full Hessian). This approach is inspired from recent work on gradient-based methods for finding SOSPs [10, 56] and effectively decouples *negative curvature estimation* from progress in the large gradient setting. As a result, RSPI does not suffer from the exponential scaling of the vanilla random search approach : we show that the overall complexity of finding a SOSP in terms of function evaluations becomes *linear* in the problem dimension.

- Finally, we verify our results empirically and compare our novel algorithm to standard random-search methods. Our results show improvements of RSPI both in terms of algorithm iterations and (crucially) wall-clock time.

## 2   Related work

**Direct-search vs methods that approximate gradients**   A wide variety of algorithms enter the class of DFO methods. One common distinction is made according to whether or not the algorithm explicitly computes a gradient approximation. Direct-search (DS) and pattern-search (PS) methods only rely on function evaluations to validate a step along a direction, sampled according to some scheme. In contrast, a second group of methods explicitly compute a gradient approximation [48, 23, 12]. Most methods in the latter group rely only on first-order information, except [58] that incorporates some second-order information to compute an estimate of the gradient. However, their approach focuses on convex optimization and therefore does not discuss second-order stationarity. Instead, [55] showed that $\mathcal{O}(\epsilon^{-2})$ approximate gradient computations are enough to reach an $\epsilon-$SOSP. Unlike DS, the gradient-free method analyzed in [55] computes an approximation of the gradient and, as a result, it matches the convergence rate guarantees of their exact gradient-based counterparts (up to constants). However, existing lower bounds clearly show that DS methods have a worst rate of convergence. Of special interest to us is the complexity w.r.t. the dimension of the problem, which is exponential, see paragraph "Lower bounds" below.

**First-order guarantees of DS.**   A recently proposed variant of DS is the stochastic-three-points (STP) approach proposed in [6] that simply samples a direction at random on the sphere and accepts the step if it decreases the function. This is, in some sense, a simpler variant of direct search compared to adaptive methods such as [54] that also include an adaptive mechanism of the step-size selection to ensure convergence without having to rely on knowing certain quantities such as the smoothness constant of the objective. As shown in [6], finding a point $\mathbf{x}$ such that $\|\nabla f(\mathbf{x})\| \le \epsilon$ with STP requires $\mathcal{O}(d\epsilon^{-2})$ function evaluations. If instead the perturbation directions are sampled from a fixed basis and not from the unit sphere, the best known complexity increases to $\mathcal{O}(d^2\epsilon^{-2})$ [54, 6]. Ignoring the dependency on the dimension, these results match the iteration complexity of steepest descent [47].

**Second-order guarantees for PS and DS.**   Second-order convergence guarantees have been developed in the context of (generalized) pattern search (GPS) methods, which share similarities with direct search methods. These methods sample update directions from a positive spanning set $\mathcal{D}$. The work of [1] proved a type of "pseudo-second-order" stationarity condition for GPS, where

the Hessian is positive semidefinite in the directions of the basis $\mathcal{D}$ (but not with respect to all vectors in the space). A variant of GPS that explicitly constructs an approximate Hessian was shown by [2] to converge to second-order stationary points under some assumptions on the quality of the approximation. The results discussed so far typically consider the properties of the limit points of the sequence of iterates and do not provide worst-case complexity rates of convergence. In contrast, [26] proved convergence to a second-order stationary point as well as derived worst-case guarantees (upper bounds only) for a variant of a direct search approach that constructs an approximate Hessian matrix using a finite difference approach.

**Model-based approaches.** Model-based methods construct a function approximation which is used to compute the next step and updated at every iteration. The literature on such methods is broad and we refer the reader to [38] for a survey of the relevant work. To the best of our knowledge, these methods also require an approximation of the Hessian to obtain second-order guarantees.

**Lower bounds.** [59, 24] show that random search suffers from an exponential dependency to the dimension of the problem. Similar results are discussed in [20] for derivative-free methods that use two function evaluations to approximate derivatives. These results serve as a motivation for this work to improve the complexity of random search methods in terms of the input dimension. We mostly focus on designing a new type of random search that achieves better worst-case guarantees than existing lower bounds for vanilla random search.

**Inexact Power Iteration.** Computing the largest eigenvector of a matrix has many applications in statistics and data analysis. Iterative methods such as the power iteration and the Lanczos algorithm are commonly used to solve this problem [25]. Perhaps the most complete and up-to-date analysis of convergence of inexact power methods for eigenspace estimation can be found in [5, 29]. Crucially, these convergence rates depend on the eigenvalue distribution. If one instead only seeks a direction aligned with a suboptimal but large eigenvalue, the rate becomes independent of the eigenvalue distribution [37]. Alternatively to the power method, stochastic methods such as Oja's algorithm [49] benefit from a cheaper iteration cost. Improvements have been proposed in [52] that analyzes a variance reduced method that achieves a linear rate of convergence. An accelerated variant of stochastic PCA is also proposed in [18].

## 3 Analysis

We work in $\mathbb{R}^d$ with the standard Euclidean norm $\|\cdot\|$. Our goal is to optimize a twice continuously differentiable non-convex function $f(\cdot) : \mathbb{R}^d \to \mathbb{R}$ without having access to gradients. We need the following assumption, standard in the literature on SOSPs [33], also in the DFO setting [55].

**Assumption 1.** *The function $f(\cdot)$ is lower bounded, $L_1$-smooth and $L_2$-Hessian Lipschitz.*

In line with many recent works on non-convex optimization [32, 35, 34, 16, 10, 55, 4], our goal is to find an $(\epsilon, \gamma)$-second-order stationary point (SOSP), i.e. a point $\mathbf{x}$ such that:

$$\|\nabla f(\mathbf{x})\| \leq \epsilon \text{ and } \lambda_{\min}(\nabla^2 f(\mathbf{x})) \geq -\gamma, \tag{1}$$

with $\epsilon, \gamma > 0$. We analyze two different DFO algorithms: (1) a two-step random search method (RS, Algorithm 1) – similar to STP [6] and (2) a novel random search method (RSPI, Algorithm 2) that extracts negative curvature via Algorithm 3. We show that, while the approximation of negative eigendirections requires extra computation, the overall complexity of RSPI to find a SOSP is much lower than RS *in terms of number of function evaluations*.

**Theorem 1** (Main result, informal)**.** *The complexity — in terms of number of function evaluations — of vanilla Random Search (similar to STP, Algorithm 1) for finding second-order stationary points depends **exponentially** on the problem dimension (see Lemma 3 and Theorem 5).*
*This dependency can be reduced to **linear** by computing an approximate negative curvature direction at each step (i.e. Algorithm 2, see Theorem 10).*

We will use $\mathcal{O}(\cdot), \Theta(\cdot), \Omega(\cdot)$ to hide constants which do not depend on any problem parameter.

### 3.1 Two-step Random Search

We analyze a variant of random search that uses a strategy consisting of two steps designed to maximally exploit gradient and curvature. We will use this variant as a surrogate for vanilla random searches such as STP [6] that only use one step. We note that the convergence rate of the latter

approach does not theoretically outperform the two-step approach since the gradient step is the same in both approaches. Algorithm 1 shows that the two-step random search samples two symmetric perturbations from a sphere in $\mathbb{R}^d$ with some radius, and updates the current solution approximation $\mathbf{x}_k$ if one of the two perturbations decreases the objective function. The sampling procedure is repeated twice at each iteration, using different sampling radii $\sigma_1$ and $\sigma_2$. One sampling radius ($\sigma_1$) is tuned to exploit the gradient of the objective function while the other ($\sigma_2$) is tuned to exploit negative curvature. One could also interpret the two steps as a one step algorithm with an adaptive step-size, although we explicitly write down the two steps for pedagogic reasons. Note that, in our analysis, the values $\sigma_1$ and $\sigma_2$ are constant, although we will see that in practice, decreasing values can be used, as is typically the case with step-sizes in gradient-based methods (see experiments). We divide the analysis into two cases: one for the case when the gradient is large and the other when we are close to a strict saddle (i.e. we still have negative curvature to exploit).

---

**Algorithm 1** TWO-STEP RANDOM SEARCH (RS).

Similar to the STP method [6], but we alternate between two perturbation magnitudes: $\sigma_1$ is set to be optimal for the large gradient case, while $\sigma_2$ optimal to escape saddles.

1: **Parameters** $\sigma_1, \sigma_2 > 0$ (see Theorem 5)
2: Initialize $\mathbf{x}_0$ at random
3: **for** $k = 0, 2, 4, \cdots, 2K$ **do**
4:    $\mathbf{s}_1 \sim \mathcal{S}^{d-1}$ (uniformly)
5:    $\mathbf{x}_{k+1} = \arg\min\{f(\mathbf{x}_k), f(\mathbf{x}_k + \sigma_1 \mathbf{s}_1), f(\mathbf{x}_k - \sigma_1 \mathbf{s}_1)\}$            *# Gradient step*
6:    $\mathbf{s}_2 \sim \mathcal{S}^{d-1}$ (uniformly)
7:    $\mathbf{x}_{k+2} = \arg\min\{f(\mathbf{x}_{k+1}), f(\mathbf{x}_{k+1} + \sigma_2 \mathbf{s}_2), f(\mathbf{x}_{k+1} - \sigma_2 \mathbf{s}_2)\}$      *# Curvature step*
8:    Optional: Update $\sigma_1$ and $\sigma_2$ (see experiments)
9: **end for**

---

**Case 1: Large gradients.** First we consider the case $\|\nabla f(\mathbf{x})\| \geq \epsilon$. Under this assumption, the stepsize can be tuned [2] to yield a decrease proportional to $\epsilon^2/d$.

---

**Lemma 2.** *Let $f(\cdot)$ be $L_1$-smooth and $\|\nabla f(\mathbf{x}_k)\| \geq \epsilon$. Algorithm 1 with $\sigma_1 = \epsilon/(L_1\sqrt{2\pi d})$ yields $\mathbf{E}[f(\mathbf{x}_{k+1}) - f(\mathbf{x}_k)|\mathbf{x}_k] \leq -\Omega\left(\frac{\epsilon^2}{L_1 d}\right)$, where $\mathbf{E}[\cdot|\mathbf{x}_k]$ denotes the conditional expectation w.r.t. $\mathbf{x}_k$.*

---

The proof follows directly from Lemma 3.4 in [6] and is presented in the appendix.

**Case 2: Close to a strict saddle.** We now address the case where $\|\nabla f(\mathbf{x})\| \leq \epsilon$ but $\lambda_{\min}(\nabla^2 f(\mathbf{x})) \leq -\gamma$, with $\gamma, \epsilon > 0$. Similarly to the analysis of gradient-based methods [33, 39], our approach consists in first approximating $f(\mathbf{x})$ around $\mathbf{x}$ with a quadratic $\tilde{f}(\mathbf{x})$, and then bounding the error on the dynamics. Hence, our first task is to estimate the probability of having a decrease in function value with a single step of random search around a *quadratic saddle*.

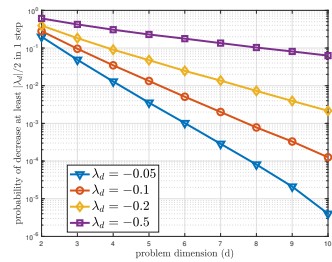

---

**Lemma 3** (Curse of dimensionality of RS around a saddle).
*Consider a $d$-dimensional ($d \geq 4$) quadratic saddle $\tilde{f}(\cdot)$ centered at the origin with eigenvalues $\lambda_1 \geq \cdots \geq \lambda_d$, with $\lambda_d < 0$. Set $\gamma := |\lambda_d|$ and $L_1 := \max\{\lambda_1, |\lambda_d|\}$ (cf. definition SOSP in Equation 1). Starting from the origin, a random step $\mathbf{s}_2 \sim \mathcal{S}^{d-1}(\sigma_2)$ is such that*

$$\Pr\left[\tilde{f}(\mathbf{s}_2) - \tilde{f}(\mathbf{0}) \leq -\frac{\gamma}{2}\sigma_2^2\right] \geq \left(\frac{\gamma}{4L_1}\right)^{\frac{d}{2}}. \quad (2)$$

*Moreover, if $\lambda_1 = \lambda_2 = \cdots = \lambda_{d-1} > 0$ (worst case scenario), we also have*

$$\Pr\left[\tilde{f}(\mathbf{s}_2) - \tilde{f}(\mathbf{0}) \leq -\frac{\gamma}{2}\sigma_2^2\right] \leq \mathcal{O}\left(2^{-d}\sqrt{d}\right). \quad (3)$$

Figure 1: Behavior of a step of vanilla random search on a quadratic saddle centered at the origin. The Hessian has $d - 1$ positive eigenvalues equal to 1 and one negative eigenvalue equal to $\lambda_d$. Plotted is the probability of a decrease ($1e6$ runs) of at least $|\lambda_d|/2$ starting from the origin ($\sigma_2 = 1$). Performance degrades exponentially with the problem dimension, as predicted by Lemma 3.

---

[2]Such tuning is optimal, please refer to the proof for details.

The proof is presented in the appendix and is based on arguments on the geometry of high-dimensional spheres: finding an escape direction at random becomes increasingly hard as the dimension increases.

Lemma 3 provides both an upper and a lower bound on the expected[3] number of function evaluations needed by vanilla random search to escape a saddle. As it is the case for vanilla gradient descent [19], the complexity grows exponentially with the dimension. We emphasize that the exponential dependency itself is not a new result (see Section 2) and that it can be extended to hold globally (not only for one step) [59]. We will use the upper bound of Lemma 3 to derive a new result in Theorem 5 that proves convergence to a *second-order* stationary point.

Further, we note that, for the bounds in Lemma 3, the decrease in function value is proportional to $\sigma_2^2$ — yet the decrease probability is independent from $\sigma_2^2$. This might seem unintuitive at first, but it is simply due to the radially isotropic structure of quadratic landscapes. However, the landscape around $\mathbf{x}_k$ is only approximately quadratic : under Assumption 1, we have (for a proof please refer to [39]) that $|f(\mathbf{y}) - \tilde{f}(\mathbf{x})| \leq \frac{L_2}{2}\|\mathbf{y} - \mathbf{x}\|^3$ for all $\mathbf{x}, \mathbf{y} \in \mathbb{R}^d$. Hence, in order to lift the bound to the non-quadratic case we need to consider "small" perturbations $\sigma_2 = \mathcal{O}(1/L_2)$, so that the quadratic approximation remains valid. The proof of the next result can be found in the appendix.

**Lemma 4.** *Let $f(\cdot)$ be $L_1$-smooth and $L_2$-Hessian-Lipschitz. Assume $\|\nabla f(\mathbf{x}_k)\| \leq \epsilon$ and $\lambda_{\min}(\nabla^2 f(\mathbf{x}_k)) \leq -\gamma = -\epsilon^{2/3}$. Then Algorithm 1 with $\sigma_2 = \frac{\epsilon^{2/3}}{2L_2}$ is s.t.*

$$\mathbf{E}[f(\mathbf{x}_{k+1}) - f(\mathbf{x}_k)|\mathbf{x}_k] \leq -\Omega\left(\left(\frac{\gamma}{4L_1}\right)^{\frac{d}{2}}\epsilon^2\right). \tag{4}$$

With no surprise, we see that the exponential complexity already found for the quadratic case in Lemma 3 directly influences the magnitude of the per-iteration decrease. We remark that the choice $\gamma = \epsilon^{2/3}$ is optimal for our proof, based on Hessian smoothness (see proof of Lemma 4 & 7).

**Joint analysis.** We define the following sets that capture the different scenarios:

$$\mathcal{A}_1 = \{\mathbf{x} : \|\nabla f(\mathbf{x})\| \geq \epsilon\} \, ; \, \mathcal{A}_2 = \{\mathbf{x} : \|\nabla f(\mathbf{x})\| < \epsilon \text{ and } \lambda_{\min}(\nabla^2 f(\mathbf{x})) \leq -\gamma\} \, .$$

By Lemma 2 and 4, we have that for any $\mathbf{x}_k \in \mathcal{A}_1 \cup \mathcal{A}_2$, Equation 4 holds — the bottleneck scenario being $\mathcal{A}_2$. Since under Assumption 1 the function is lower bounded and Algorithm 1 is a descent method, we get convergence in a finite number of steps to an SOSP.

**Theorem 5.** *Fix the desired level of accuracy $\epsilon > 0$. Assume $f(\cdot)$ satisfies Assumption 1. Two-step RS (Alg. 1) with parameters $\sigma_1 = \frac{\epsilon}{L_1\sqrt{2\pi d}}$ and $\sigma_2 = \frac{\epsilon^{2/3}}{2L_2}$ returns in expectation an $(\epsilon, \epsilon^{2/3})$-second-order stationary point in $\mathcal{O}\left(\kappa^d \epsilon^{-2}\right)$ function evaluations, where $\kappa = \Theta(L_1/\gamma) = \Theta(L_1\epsilon^{-2/3})$.*

## 3.2 Power Iteration Random Search

The purpose of this section is to modify vanilla random search to reduce the exponential dependency of the convergence rate on the problem dimension, illustrated by the lower bound in Lemma 3. We propose a new type of random search that is coupled with a derivative-free routine (named DFPI) in order to overcome the curse of dimensionality. We note that this type of "hybrid" method is not completely new in the literature, see e.g. [26], but in contrast to prior work, the computational cost of the DFPI routine in terms of the input dimension is low. This is achieved by computing an approximation of the eigenvector corresponding to the most negative eigenvalue of the Hessian based on a noisy power method [29], which is inspired from recent works on Hessian-free negative curvature exploitation techniques in gradient-based optimization [41, 10]. The resulting method is shown as Algorithm 2 and the DFPI routine is presented as Algorithm 3.

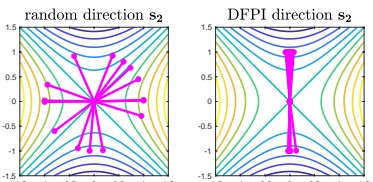

Figure 2: Difference between the two procedures to generate $\mathbf{s}_2$ when initialized at a quadratic saddle (blue = small values). Each magenta point is a random update direction $\mathbf{s}_2 \sim \mathcal{S}^{d-1}$ (left) or the result of the DFPI procedure with 3 iterations (right).

---

[3]Let $V$ be an event that happens in a trial with probability $p$. Then the expected number of trials to first occurrence of $V$ is $1/p$.

We highlight that RSPI does not require computation and storage of the full Hessian (in contrast to AHDS that performs $d^2$ function calls). We instead approximate only the leading eigenvector using approximate zero-order Hessian-vector products ($\approx d \log d$ function calls: Lemma 9). This provides a large speed-up in high dimensions, see wall-clock times reported later.

---

**Algorithm 2** RANDOM SEARCH PI (RSPI).

Same structure as Alg. 1, only difference is in line 6: the curvature exploitation step is based on a perturbation sampled from a non-isotropic distribution aligned with negative eigendirections, computed by Alg. 3 (see Fig.2).

1: **Parameters** $\sigma_1, \sigma_2 > 0$ (see Thm 10)
2: Initialize $\mathbf{x}_0$ at random
3: **for** $k = 0, 2, 4, \cdots 2K$ **do**
4: $\quad \mathbf{s}_1 \sim \mathcal{S}^{d-1}$ (uniformly)
5: $\quad \mathbf{x}_{k+1} = \arg\min\{f(\mathbf{x}_k), f(\mathbf{x}_k + \sigma_1 \mathbf{s}_1), f(\mathbf{x}_k - \sigma_1 \mathbf{s}_1)\}$
6: $\quad \mathbf{s}_2 = \text{DFPI}(\mathbf{x}_k) \leftarrow$ Algorithm 3
7: $\quad \mathbf{x}_{k+2} = \arg\min\{f(\mathbf{x}_{k+1}), f(\mathbf{x}_{k+1} + \sigma_2 \mathbf{s}_2), f(\mathbf{x}_{k+1} - \sigma_2 \mathbf{s}_2)\}$
8: $\quad$ Optional: Update $\sigma_1$ and $\sigma_2$ (see experiments)
9: **end for**

---

**Algorithm 3** DERIVATIVE-FREE POWER ITERATION (DFPI)

A noisy derivative-free power method to approximate negative curvature directions, as shown in Lemma 6. Every iteration of DFPI requires $4d$ function evals, hence the routine requires $K_{\text{DFPI}} = 4d T_{\text{DFPI}}$ function evals overall.

1: **Parameters** $c, r, \eta > 0$ and $T_{\text{DFPI}} \in \mathbb{Z}^+$ (see Theorem10)
2: **INPUTS :** $\mathbf{x} \in \mathbb{R}^d$,
3: $\mathbf{s}_2^{(0)} \sim \mathcal{S}^{d-1}$ (uniformly)
4: **for** $t = 0 \dots T_{\text{DFPI}} - 1$ **do**
5: $\quad$ Set $\mathbf{g}_+ = \sum\limits_{i=1}^{d} \frac{f(\mathbf{x}+r \cdot \mathbf{s}_2^{(t)}+c \cdot \mathbf{e}_i) - f(\mathbf{x}+r \cdot \mathbf{s}_2^{(t)}-c \cdot \mathbf{e}_i)}{2c} \mathbf{e}_i$
6: $\quad$ Set $\mathbf{g}_- = \sum\limits_{i=1}^{d} \frac{f(\mathbf{x}-r \cdot \mathbf{s}_2^{(t)}+c \cdot \mathbf{e}_i) - f(\mathbf{x}-r \cdot \mathbf{s}_2^{(t)}-c \cdot \mathbf{e}_i)}{2c} \mathbf{e}_i$
7: $\quad$ **Update:** $\mathbf{s}_2^{(t+1)} = \mathbf{s}_2^{(t)} - \eta \frac{\mathbf{g}_+ - \mathbf{g}_-}{2r}$
8: $\quad$ Normalize $\mathbf{s}_2^{(t+1)} = \mathbf{s}_2^{(t+1)}/\|\mathbf{s}_2^{(t+1)}\|$
9: **end for**
10: **RETURN :** $\mathbf{s}_2^{(T_{\text{DFPI}})}$

---

As in the last subsection, we split the analysis into two cases. First, in the case of large gradients, i.e. $\|\nabla f(\mathbf{x})\| \geq \epsilon$, we can simply re-use the result of Lemma 2. The second case (sufficient negative curvature) requires to show that the vector $\mathbf{s}_2$ returned by the DFPI procedure (Alg. 3) yields a decrease of the objective.

**Exploiting negative curvature close to a saddle.** We consider again the case where $\|\nabla f(\mathbf{x}_k)\| \leq \epsilon$ and $\lambda_{\min}(\nabla^2 f(\mathbf{x}_k)) \leq -\gamma$. We saw in Lemma 3, that isotropic sampling in the neighborhood of $\mathbf{x}_k$ provides a function decrease only after an exponential number of iterations, in the worst case. We show that if the perturbation is instead sampled from a distribution which satisfies the next assumption, the number of required iterations drastically decreases.

**Assumption 2.** *Consider* $\mathbf{x}$ *s.t.* $\lambda_{\min}(\nabla^2 f(\mathbf{x})) \leq -\gamma$. *The direction* $\mathbf{s}_2$, *output of DFPI after* $K_{DFPI}$ *function evaluations, returns in expectation a good approximation to the most negative eigenvalue of* $\nabla^2 f(\mathbf{x})$. *Specifically,* $\mathbf{E}[\mathbf{s}_2^\top \nabla^2 f(\mathbf{x}) \mathbf{s}_2] \leq \lambda_{\min}(\nabla^2 f(\mathbf{x})) + \frac{\gamma}{2}$.

This assumption — *which we will soon formally validate* — is motivated by the fact that DFPI is an approximate (noisy) power method on $\mathbf{A}(\mathbf{x}) := \mathbf{I} - \eta \nabla^2 f(\mathbf{x})$, hence can be used to estimate the maximum eigenvalue of $\mathbf{A}(\mathbf{x})$ — which is minimum eigenvalue of $\nabla^2 f(\mathbf{x})$ if $\eta \leq 1/L_1$.

**Lemma 6.** *Let* $f(\cdot)$ *be* $L_1$-*smooth and* $L_2$-*Hessian-Lipschitz. The iteration of DFPI can be seen as a step of a noisy power method:* $\mathbf{s}_2^{(t+1)} = (\mathbf{I} - \eta \nabla^2 f(\mathbf{x})) \mathbf{s}_2^{(t)} + \boldsymbol{\xi}_{DFPI}^{(t)}$, *with* $\|\boldsymbol{\xi}_{DFPI}^{(t)}\| = \mathcal{O}(rL_2 + \frac{c}{r} L_1 \sqrt{d})$. *In particular,* $\|\boldsymbol{\xi}_{DFPI}^{(t)}\| \to 0$ *as* $r, \frac{c}{r} \to 0$; *hence the error can be made as small as needed within the limits of numerical stability. In addition, if* $f(\cdot)$ *is quadratic, we have* $\|\boldsymbol{\xi}_{DFPI}^{(t)}\| = 0$.

The proof is presented in the appendix. The bound on $\boldsymbol{\xi}_{\text{DFPI}}^{(t)}$ is enough for us to apply the well-known convergence rates for the noisy power method [29] and to motivate a $\mathcal{O}(\log(d))$ bound on the DFPI iterations needed to satisfy Assumption 2. Before diving into this, we show that directly using Assumption 2 actually makes the iteration[4] complexity dimension independent.

**Lemma 7.** *Let $f(\cdot)$ be $L_1$-smooth and $L_2$-Hessian-Lipschitz, and assume $\|\nabla f(\mathbf{x}_k)\| \leq \epsilon$ and $\lambda_{\min}(\nabla^2 f(\mathbf{x}_k)) \leq -\gamma = -\epsilon^{2/3}$. Under Assumption 2, RSPI (Algorithm 2) with $\sigma_2 = \frac{\gamma}{2L_2}$ (choice as Theorem 5) yields $\mathbf{E}[f(\mathbf{x}_{k+1}) - f(\mathbf{x}_k)|\mathbf{x}_k] \leq -\Omega(\epsilon^2)$, independent of the problem dimension.*

**Iteration complexity.** We now combine Lemma 2 and 7.

**Proposition 8.** *Fix the desired level of accuracy $\epsilon > 0$. Assume $f(\cdot)$ satisfies Assumptions 1 and 2. RSPI (i.e. Alg. 2) with parameters $\sigma_1 = \frac{\epsilon}{L_1\sqrt{2\pi d}}$ and $\sigma_2 = \frac{\epsilon^{2/3}}{2L_2}$ (same choice as Theorem 5) returns in expectation an $(\epsilon, \epsilon^{2/3})$-second-order stationary point in $\mathcal{O}(\epsilon^{-2})$ iterations.*

**Overall number of function evaluations.** While Proposition 8 shows that the number of RSPI iterations is, conditioned on Assumption 2, independent of the problem dimension, it hides the number of function evaluations needed for the assumption to hold. To include this into the final complexity (Theorem 10) – *i.e. to drop Assumption 2* – we need a bound for convergence of noisy power methods [29].

**Lemma 9** (Consequence of Corollary 1.1 in [29])**.** *Let the parameters of DFPI be such that the error $\boldsymbol{\xi}_{DFPI}$ is small enough (always possible, as shown in Lemma 6). Let $\eta \leq 1/L_1$. Let $\gamma = \epsilon^{2/3}$; for a fixed RSPI iteration, $T_{DFPI} = \mathcal{O}\left(\epsilon^{-2/3}L_1 \log\left(\frac{d}{\delta^2}\right)\right)$ DFPI iterations are enough to ensure validity of Assumption 2 at $\mathbf{x}_k$ with probability $1 - \delta^{\Omega(1)} - e^{\Omega(d)}$.*

A derivation is included for completeness in the appendix. We are ready to state the main result.

**Theorem 10.** *Fix the accuracy $\epsilon > 0$. Assume $f(\cdot)$ satisfies Assumption 1. Algorithm 2 with parameters $\sigma_1 = \frac{\epsilon}{L_1\sqrt{2\pi d}}$ and $\sigma_2 = \frac{\epsilon^{2/3}}{2L_2}$ (same choice as Theorem 5) combined with DFPI (Alg. 3) with parameters $\eta \leq 1/L_1$, $T_{DFPI} = \mathcal{O}\left(\epsilon^{-2/3}L_1 \log(d)\right)$ and $c, r$ sufficiently small, returns in expectation an $(\epsilon, \epsilon^{2/3})$-second-order stationary point in $\mathcal{O}\left(\epsilon^{-8/3}d \log(L_1 d)\right)$ function evaluations.*

*Proof.* Fix $\delta$ s.t. the probability of success $p := 1 - \delta^{\Omega(1)} - e^{-\Omega(d)} = \mathcal{O}(1)$ in Lemma 9 is positive. The number of function evaluations needed for Lemma 9 to hold in expectation at each step (i.e. the burden of Assumption 2) is then $K_{\text{DFPI}}/p = \mathcal{O}\left(\epsilon^{-2/3}L_1 d \log(d)\right)$. We conclude by Proposition 8. $\qquad\square$

**Significance and Novelty of the analysis.** To the best of our knowledge, the result in Theorem 10 is the first to prove convergence of a type of random search to a *second-order* stationary point with a *linear* dependency to the input dimension. Our experimental results (section 4) confirm this significant speed-up is observed in practice. The analysis presented in appendix relies on some new geometric arguments for high-dimensional spaces coupled with more classical optimization bounds. We again emphasize that the analysis technique is different from gradient-free methods that approximate the gradients. For instance [55] define the error between the approximate gradient $q(x_k, h_k)$ and the exact gradient $\nabla f(x_k)$ as $\epsilon_k = q(x_k, h_k) - \nabla f(x_k)$. They then require this error to be bounded, i.e. $|\epsilon_k| \leq c_h|h|$ where $c_h$ is a constant and $h$ is the size of the step used in the finite difference formula. The constant $c_h$ is directly controlled by $h$ and can be made as small as possible to better approximate the result of the gradient-based method. In our case, the update direction is sampled at random, and we therefore have to rely on a different proof technique that involves probabilistic geometric arguments in high-dimensional spaces.

---

[4]The dependency on the dimension will however show up in the final number of function evaluations (Theorem 10). However, such dependency is not exponential even in the worst case.

**How to further speed up DFPI with SPSA.**    Each DFPI iteration requires $4d$ function evaluations. While this complexity is necessary to build an arbitrarily good finite difference (FD) approximation of second order information (needed by Lemma 9), in practice a more rough estimate of Hessian-vector products can be obtained using cheap randomized techniques. In particular, in the experiments in the next section, we show that an SPSA estimator [53] of $\mathbf{g}_+$ and $\mathbf{g}_-$ is in fact sufficient to achieve acceleration in the performance with respect to the two-step random search. In particular, SPSA computes $\mathbf{g}_+, \mathbf{g}_-$ as $\mathbf{g}_\pm = \sum_{i=1}^{d} \frac{f(\mathbf{x} \pm r\mathbf{s}_2^{(t)} + c\boldsymbol{\Delta}) - f(\mathbf{x} \pm r\mathbf{s}_2^{(t)} - c\boldsymbol{\Delta})}{2c\Delta_i} \mathbf{e}_i$, where $r, c > 0$ and $\boldsymbol{\Delta}$ is a vector of $d$ random variables, often picked to be symmetrically Bernoulli distributed. SPSA is asymptotically unbiased and only requires $4$ function calls, as opposed to the $4d$ needed by FD. However, the variance of SPSA does not arbitrarily decrease to zero as $c, r$ vanish, as it is instead the case for FD: it saturates [53]. As a result, Lemma 9 would not always hold. In the appendix, we provide an extensive comparison between FD and SPSA for DFPI: we show that, for the sake of Assumption 2, the error in the SPSA estimator is acceptable for our purposes, for small enough $\eta$ (see use in Alg. 3).

## 4   Experiments

In this section we verify our theoretical findings empirically. Specifically, we set two objectives: (1) to evaluate the performance RS and RSPI, to verify the validity of Theorem 1, and (2) to compare the performance of these two methods against existing random search methods. For the latter, we run the two-step Random Search (RS) and Random Search Power Iteration (RSPI) against the Stochastic Three Points (STP) method, the Basic Direct Search (BDS) [54] and the Approximate Hessian Direct Search (AHDS). We recall that AHDS explicitly constructs an approximation to the Hessian matrix in order to extract negative curvature. Descriptions of each algorithm are provided in the appendix.

**Setup.**    All experiments follow a similar procedure. In each task, all algorithms are initialized at a strict saddle point and executed for the same number of iterations. We report the optimality gap as a function of iterations and wall-clock time. Additionally, we report the norm of the gradient vector in the appendix. Since all algorithms are stochastic, the experimental process is repeated multiple times (wherever possible using a different saddle point as initialization) and the results are averaged across all runs. For each task, the hyperparameters of every method are selected based on a coarse grid search refined by trial and error. For RS and RSPI the parameters $\sigma_1$ and $\sigma_2$ are initialized and updated in the same manner, hence the only difference between the two is that RSPI extracts negative curvature explicitly whereas the two-step RS samples a direction at random (see Figure 2). We choose to run DFPI for 20 iterations for all the results shown in the paper. Empirically, we observed that performing more iterations does not further improve the overall performance of RSPI. The hyperparameters used for each method are provided in the appendix and the code for reproducing the experiments is available online[5].

**Function with growing dimension.**    We start by considering the following benchmarking function (see App. D for an illustration)
$$f(x_1, \cdots, x_d, y) = \frac{1}{4} \sum_{i=1}^{d} x_i^4 - y \sum_{i=1}^{d} x_i + \frac{d}{2} y^2, \tag{5}$$
which has a unique strict saddle point (i.e. with negative curvature) at $P1 = (0, \cdots, 0, 0)$ and two global minima at $P2 = (1, \cdots, 1, 1)$ and $P3 = (-1, \cdots, -1, -1)$. The results in Fig. 3 illustrate that both the two-step RS method as well as the RSPI algorithm are able to consistently escape the saddle across all dimensions. While in low-dimensional settings ($d = 5$) the RSPI algorithm is outperformed by the two-step RS and the AHDS in terms of their behavior as a function of run-time, the situation is clearly reversed as the dimensionality grows. For $d = 100, 200$ the two-step RS achieves progress at a very slow rate and a high number of iterations is needed in order to achieve convergence to a second-order stationary point. In contrast, RSPI successfully approximates the negative curvature of the objective to efficiently escape the saddle point, allowing the algorithm to achieve progress at a faster rate. The fact that for $d = 5$ the AHDS algorithm requires less time than RSPI to converge to a second-order stationary point, indicates that in low-dimensional settings the cost incurred by the power iterations within RSPI is higher than the cost of approximating the entire Hessian matrix. However, for higher values of $d$, AHDS quickly becomes inefficient and expensive. Further, STP performs worse than RS, simply because it employs only one sampling radius (RS uses two).

---

[5] https://github.com/adamsolomou/second-order-random-search

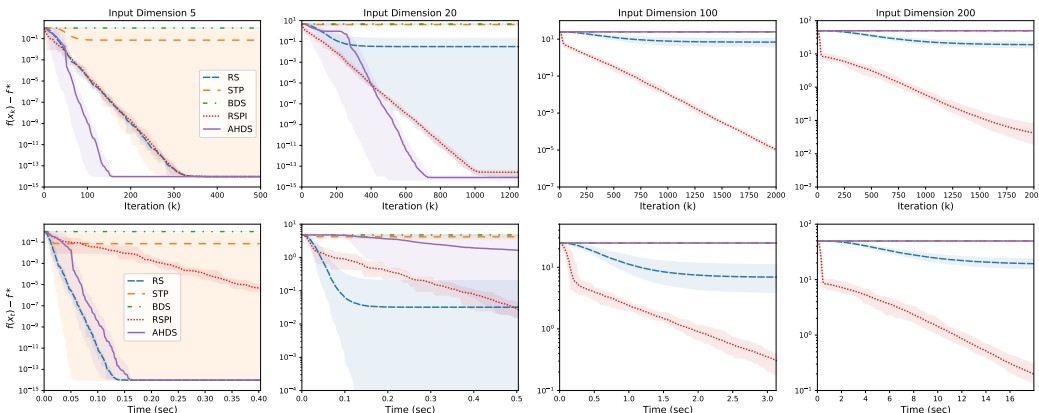

Figure 3: Performance while minimizing the objective in Eq. (5) for different $d$. Confidence intervals show min-max intervals over ten runs. All algorithms are initialized at the strict saddle point across all runs. For $d = 100, 200$, the lines for STP, BDS and AHDS overlap each other as none of the methods achieve progress in terms of function value.

**Rastrigin function.** Next, we conduct experiments on the Rastrigin function, a popular benchmark in the literature [28]. For any $\mathbf{x} \in \mathbb{R}^d$, the $d$-dimensional Rastrigin function is defined as

$$f(\mathbf{x}) = 10d + \sum_{i=1}^{d} (x_i^2 - 10\cos(2\pi x_i)).\qquad(6)$$

The function has a unique global minimizer at $\mathbf{x}^* = \mathbf{0}$, whereas the number of stationary points (including strict saddles) grows exponentially with $d$. Based on Lemma 3, we expect that having a single direction of negative curvature will challenge the core mechanism of each algorithm while trying to escape the saddle. To that end, we ensure that at each initialization point there exist a single direction of negative curvature across all settings of $d$. More details about the initialization process and the implications on the results are given in the appendix.

The results in Figure 4 illustrate the strong empirical performance of RSPI, not only in comparison to the two-step RS algorithm but also against the rest of the algorithms. RSPI successfully approximates the single direction of negative curvature and escapes the saddle point after one iteration. On the contrary, the two-step RS achieves minimal progress even for low dimensional settings, whereas for $d = 20$ it requires more than 300 iterations to escape the saddle (see the gradient norm plot in the appendix). For higher dimensional settings, two-step RS does not escape at all, supporting our theoretical argument that as the problem dimension grows the probability of sampling a direction that is aligned with the direction of negative curvature decreases exponentially. Lastly, both BDS

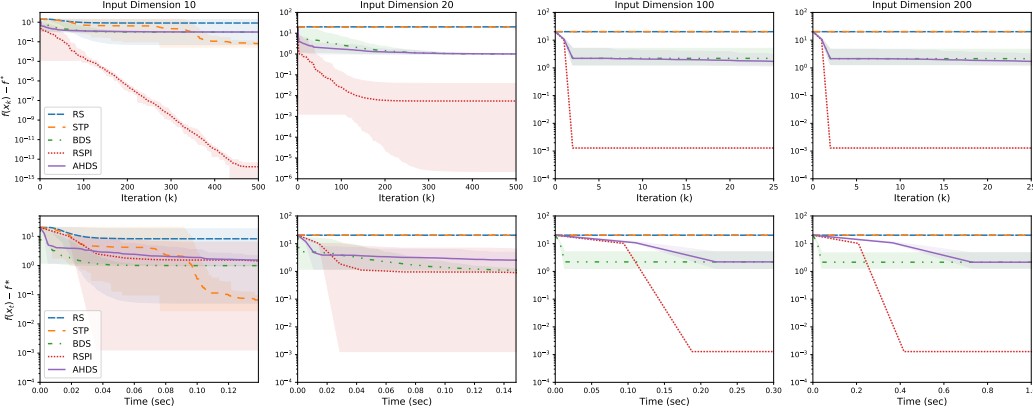

Figure 4: Optimality gap on the Rastrigin function as a function of iterations (top) and running time (bottom). Confidence intervals show min-max intervals over ten runs. All algorithms are initialized at a strict saddle point and executed for a total of $500$ iterations across all runs, however for $d = 100, 200$ no further improvement is achieved after $25$ iterations.

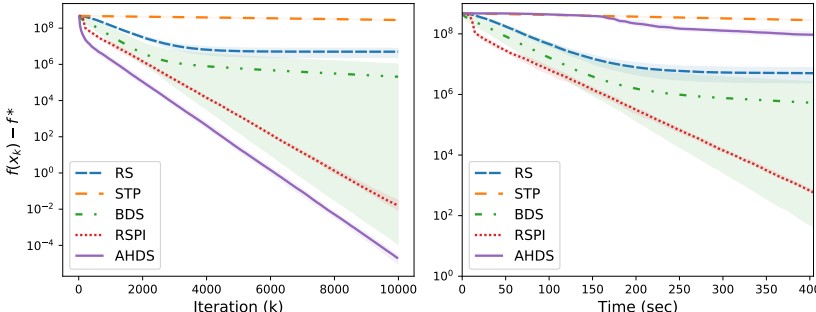

Figure 5: Empirical performance in finding the leading eigenvector of a 350-dimensional random matrix. Confidence intervals show min-max intervals over five runs. All algorithms are initialized at a strict saddle point.

and AHDS consistently escape the saddle point across all values of $d$. However, their performance remains suboptimal compared to RSPI.

**Leading eigenvector problem.** Finally, we consider the task of finding the leading eigenvector of a positive semidefinite matrix $\mathbf{M} \in \mathbb{R}^{d \times d}$. The problem is equivalent to minimizing $f(\mathbf{x}) = \|\mathbf{x}\mathbf{x}^T - \mathbf{M}\|_F^2$ [34]. Figure 5 shows the empirical performance in finding the leading eigenvector of a 350-dimensional random matrix. While at an iteration level AHDS appears to be very effective, when the iteration complexity of each method is taken into account it is the slowest to escape. Notably, a single iteration of RSPI takes (on average) 0.06 seconds, whereas a single iteration of AHDS takes approximately 9.01 seconds. This experiment clearly illustrates the computational advantages that RSPI provides while provably ensuring convergence to second order stationary points.

## 5 Conclusion

We analyzed the convergence of two types of random search methods. The first approach is a slight variation of standard random search that converges to a second-order stationary point but whose worst-case analysis demonstrates exponential complexity in terms of the function input dimension. The second random search approach we propose extracts negative curvature using function evaluations. Importantly, the dependency in terms of the function input dimension becomes linear, a result which we clearly observed in our experimental results, especially in terms of run-time.

There are a number of avenues to pursue as future work. To start off, (1) a simple extension would be to allow DFPI to store multiple candidates for negative eigendirections. As discussed in [45] and formally shown in [27], this can directly boost performance. Similarly (2) one could study the application of a zero-th order version of Neon [56]. (3) It would be then interesting to also understand if injecting noise in the process (additional exploration) can help in escaping saddles [19]. (4) Further, instead of using constant values or a predefined schedule for $\sigma_1$ and $\sigma_2$, one could analyze the commonly used adaptive strategy where these values are adapted according to whether the function is being decreased [54]. (5) We note that one could in principle relax Assumption 1 and instead work with a smoothed version of $f(\cdot)$, without requiring differentiability. (6) Finally, it would be interesting to benchmark DPFI on other machine learning problems, including for instance reinforcement learning tasks where random search methods are becoming more prevalent [43, 42]. Another potentially interesting direction would be to extend our analysis to random search methods for min-max optimization problems [3].

**Broader impact statement.** We consider our work fundamental research on the principles of learning in high dimensional systems. Hence a broader impact discussion is not applicable.

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
