# Appendix

## A    Analysis of random search (Algorithm 1)

We start by studying some properties of high dimensional spheres. We then apply these properties to show how the rate of the two-step random search (Algorithm 1) depends exponentially on the problem dimension.

### A.1    High-dimensional spheres and curse of dimensionality

We denote by $V_R(d)$ and $A_R(d)$ the volume and the surface area of the $(d-1)$ sphere with radius $R$: $\mathcal{S}^{d-1}(R) = \{\mathbf{x} \in \mathbb{R}^d \mid \|\mathbf{x}\| = R\}$. It is well known [30] that the following formulas hold:

$$A_R(d) = \frac{2\pi^{d/2}}{\Gamma\left(\frac{d}{2}\right)} R^{d-1}, \quad V_R(d) = \frac{A(d)R}{d} = \frac{2\pi^{d/2}}{d\,\Gamma\left(\frac{d}{2}\right)} R^d. \tag{7}$$

Moreover, we have the following important lemma, which can also be found in Section 1.2.4 of [30].

**Lemma 11.** *Let $\varsigma \geq 0$ and define $A_1^\varsigma(d)$ to be the surface area of the cap $\{\mathbf{x} \in \mathbb{R}^d \mid \|\mathbf{x}\| = 1, x_1 \geq \varsigma\}$, $d \geq 2$. We have:*

$$A_1^\varsigma(d) = A_1(d-1) \int_\varsigma^1 (1 - x_1^2)^{\frac{d-2}{2}} dx_1. \tag{8}$$

*Proof.* The radius of the spherical cap at height $x_1$ is $\sqrt{1 - x_1^2}$, and we have that $A_{\sqrt{1-x_1^2}}(d-1) = A_1(d-1)\left(\sqrt{1-x_1^2}\right)^{d-2}$ by the surface area formula in Equation 7. Since $A_1^\varsigma(d) = \int_\varsigma^1 A_{\sqrt{1-x_1^2}}(d-1) dx_1$, we conclude. $\square$

We will need both an upper and a lower bound on the integral above. The next result shows that both these bounds are exponential.

**Lemma 12.** *For any $\alpha > 1$,*

$$\left[\frac{1}{2}(1 - \varsigma^2)\right]^{\alpha+1} \leq \int_\varsigma^1 (1 - x^2)^\alpha dx \leq (1 - \varsigma^2)^\alpha.$$

*Proof.* The upper bound is straightforward. The lower bound in an application of Hölder's inequality (see e.g. Equation 1.1 in [13]): for a real number $p > 1$ and functions $f$ and $g$ regular enough,

$$\int_\varsigma^1 |f(x)g(x)| dx \leq \left[\int_\varsigma^1 |f(x)|^p dx\right]^{1/p} \left[\int_\varsigma^1 |g(x)|^{\frac{p}{p-1}} dx\right]^{\frac{p-1}{p}}. \tag{9}$$

Take $g$ to be constant equal to one. Then, taking everything to power $p$

$$\left[\int_\varsigma^1 |f(x)| dx\right]^p \leq (1 - \varsigma)^{p-1} \int_\varsigma^1 |f(x)|^p dx. \tag{10}$$

By applying this formula and after performing a few algebraic manipulations, we get

$$\int_\varsigma^1 (1 - x^2)^\alpha dx \geq \frac{\left(\int_\varsigma^1 (1 - x^2) dx\right)^\alpha}{(1 - \varsigma)^{\alpha-1}} = (1 - \varsigma)\left(\frac{\varsigma^3 - 3\varsigma + 2}{3(1 - \varsigma)}\right)^\alpha$$

$$= (1 - \varsigma)\left(\frac{(1 - \varsigma)^2(\varsigma + 2)}{3(1 - \varsigma)}\right)^\alpha = (1 - \varsigma)\left(\frac{1}{3}(1 - \varsigma)(\varsigma + 2)\right)^\alpha \geq \left[\frac{1}{2}(1 - \varsigma^2)\right]^{\alpha+1}, \tag{11}$$

where in the last inequality we used the fact that for $\varsigma \in [0, 1]$, $\frac{1}{3}(1 - \varsigma)(\varsigma + 2) \geq \frac{1}{2}(1 - \varsigma^2)$. $\square$

A verification of the bound above can be found in Figure 6. We note that the upper bound becomes tight as $\alpha \to \infty$, and that the lower bound becomes less pessimistic as $\varsigma \to 1$.

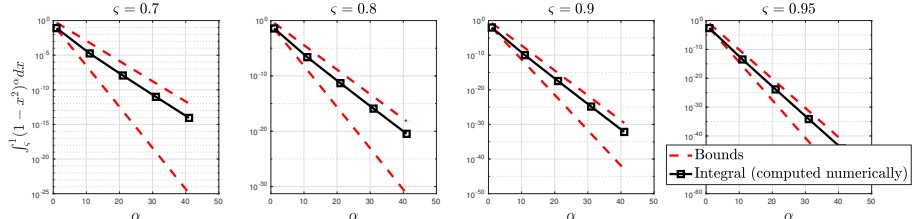

Figure 6: Numerical verification of Lemma 12. Integral computed numerically using the MATLAB `integral` function.

Putting the previous two lemmas together, we get

$$A_1(d-1)\left[\frac{1}{2}(1-\varsigma)\right]^{\alpha+1} \leq A_1^\varsigma(d) \leq A_1(d-1)(1-\varsigma^2)^\alpha, \tag{12}$$

where $\alpha = (d-2)/2$. Now we are ready to state the final lemma for high-dimensional spheres, which is verified empirically in Figure 7.

**Lemma 13** (Curse of dimensionality). *Let $\mathbf{x}$ be a random point on the surface of the unit d-ball in Euclidean space, with $d \geq 4$. For $\varsigma \in (0,1)$, we have*

$$\left[\frac{1}{2}(1-\varsigma^2)\right]^{\frac{d}{2}} \leq \Pr[|x_1| > \varsigma] \leq 2\sqrt{d-2}\left[1-\varsigma^2\right]^{\frac{d}{2}-1}. \tag{13}$$

In particular, the probability of being $\varsigma$-away from the equator **decays exponentially** with the number of dimensions.

*Proof.* The proof is just a matter of finding good upper and lower bounds on $A(d)$ as a function of $A(d-1)$, to combine with the result of Lemma 12. We are going to use the lower bound on the surface area by [30] (Equation 1.3): $A_1(d) \geq \frac{1}{\sqrt{d-2}}A_1(d-1)$. For an easy upper bound, we can instead pick $A_1(d) \leq 2A_1(d-1)$ (surface of the enclosing cylinder). Combining Lemma 11 with Lemma 12 and the bounds we just found, we get

$$\Pr[|x_1| \geq \varsigma] = \frac{A_1^\varsigma(d)}{\frac{1}{2}A_1(d)} \leq \frac{(1-\varsigma^2)^{\frac{d-2}{2}}A_1(d-1)}{\frac{1}{2\sqrt{d-2}}A_1(d-1)}, \tag{14}$$

$$\Pr[|x_1| \geq \varsigma] = \frac{A_1^\varsigma(d)}{\frac{1}{2}A_1(d)} \geq \frac{\left[\frac{1}{2}(1-\varsigma^2)\right]^{\frac{d}{2}}A_1(d-1)}{A_1(d-1)}. \tag{15}$$

$\square$

### A.2 Non-convex dynamics — the quadratic case

Here we seek to understand the behaviour of random search around a point $\mathbf{x}_k$ with negative curvature, we consider the quadratic approximation $\tilde{f}(\mathbf{x}) = f(\mathbf{x}_k) + \nabla f(\mathbf{x}_k)^\top(\mathbf{x} - \mathbf{x}_k) + \frac{1}{2}(\mathbf{x} - \mathbf{x}_k)^\top \nabla^2 f(\mathbf{x}_k)(\mathbf{x} - \mathbf{x}_k)$ where $\nabla^2 f(\mathbf{x}_k) \in \mathbb{R}^{d \times d}$. By the spectral theorem, we have $\nabla^2 f(\mathbf{x}_k) = \mathbf{V}^\top \Lambda \mathbf{V}$, where $\mathbf{V} = [\mathbf{v}_i]_{i=1}^d$, $\mathbf{v}_i \in \mathbb{R}^d$ contains an orthonormal basis of eigenvectors of $\nabla^2 f(\mathbf{x}_k)$ and $\Lambda$ is a diagonal matrix containing the eigenvalues $\lambda_1 \geq \lambda_2 \geq \cdots \geq \lambda_d$ of $\nabla^2 f(\mathbf{x}_k)$ (counted together with their multiplicity). For the setting considered in this paragraph, we have $\lambda_d < 0$.

In our first result, we consider the case $f(\mathbf{x}_k) = 0$ and $\nabla f(\mathbf{x}_k) = \mathbf{0}$.

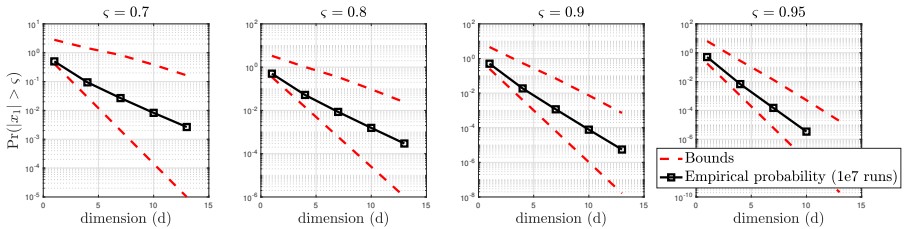

Figure 7: Numerical verification of Lemma 13. Bounds can be found in Lemma 13.

**Lemma 3** (Curse of dimensionality of RS around a saddle). *Consider a $d$-dimensional ($d \geq 4$) quadratic saddle $\tilde{f}(\cdot)$ centered at the origin with eigenvalues $\lambda_1 \geq \cdots \geq \lambda_d$, with $\lambda_d < 0$. Set $\gamma := |\lambda_d|$ and $L_1 := \max\{\lambda_1, |\lambda_d|\}$ (cf. definition SOSP in Equation 1). Starting from the origin, a random step $\mathbf{s}_2 \sim \mathcal{S}^{d-1}(\sigma_2)$ is such that*

$$\Pr\left[\tilde{f}(\mathbf{s}_2) - \tilde{f}(\mathbf{0}) \leq -\frac{\gamma}{2}\sigma_2^2\right] \geq \left(\frac{\gamma}{4L_1}\right)^{\frac{d}{2}}. \tag{2}$$

*Moreover, if $\lambda_1 = \lambda_2 = \cdots = \lambda_{d-1} > 0$ (worst case scenario), we also have*

$$\Pr\left[\tilde{f}(\mathbf{s}_2) - \tilde{f}(\mathbf{0}) \leq -\frac{\gamma}{2}\sigma_2^2\right] \leq \mathcal{O}\left(2^{-d}\sqrt{d}\right). \tag{3}$$

*Proof.* Let $x_0$ be any initial point. We seek the probability of the event

$$E_{\text{decr}} := \{\mathbf{s}_2^T \nabla^2 \tilde{f}(\mathbf{x}_0)\mathbf{s}_2 \leq -\zeta\}, \tag{16}$$

for some positive $\zeta$. First, we divide everything by $\|\mathbf{s}_2\|^2 = \sigma_2^2$, to effectively reduce the problem to the special case $\sigma_2^2 = 1$. We get

$$E_{\text{decr}} = \left\{\left(\frac{\mathbf{s}_2}{\|\mathbf{s}_2\|}\right)^T \nabla^2 \tilde{f}(\mathbf{x}_0)\frac{\mathbf{s}_2}{\|\mathbf{s}_2\|} \leq -\tilde{\zeta}\right\}, \tag{17}$$

where $\tilde{\zeta} = \zeta/\sigma_2^2$. Let us now write $\mathbf{s}_2/\|\mathbf{s}_2\|$ in the eigenbasis $\{\mathbf{v}_i\}_{i=1}^d$ of the Hessian $\nabla^2 \tilde{f}(\mathbf{x}_0)$. We have that

$$\frac{\mathbf{s}_2}{\|\mathbf{s}_2\|} = \sum_{i=1}^d a_i\mathbf{v}_i, \quad \sum_{i=1}^d a_i^2 = 1. \tag{18}$$

Hence, we can write

$$E_{\text{decr}} = \{\lambda_1 a_1^2 + \lambda_2 a_2^2 + \cdots + \lambda_d a_d^2 \leq -\tilde{\zeta}\}. \tag{19}$$

To bound the probability of this event, we construct the smaller event $E_{\text{decr}}^* \subseteq E_{\text{decr}}$:

$$E_{\text{decr}}^* := \{\lambda_1 a_1^2 + \lambda_1 a_2^2 + \cdots + \lambda_1 a_{d-1}^2 \leq |\lambda_d|a_d^2 - \tilde{\zeta}\}. \tag{20}$$

This event can be written in a reduced form, using the fact that $\sum_{i=1}^d a_i^2 = 1$; indeed

$$\lambda_1 a_1^2 + \lambda_1 a_2^2 + \cdots + \lambda_1 a_{d-1}^2 \leq |\lambda_d|a_d^2 - \tilde{\zeta} \tag{21}$$

$$\Longleftrightarrow \lambda_1 a_1^2 + \lambda_1 a_2^2 + \cdots + \lambda_1 a_{d-1}^2 + \lambda_1 a_d^2 \leq (|\lambda_d| + \lambda_1)a_d^2 - \tilde{\zeta} \tag{22}$$

$$\Longleftrightarrow \lambda_1 \leq (|\lambda_d| + \lambda_1)a_d^2 - \tilde{\zeta} \tag{23}$$

$$\Longleftrightarrow a_d^2 \geq \frac{\lambda_1 + \tilde{\zeta}}{\lambda_1 + |\lambda_d|}. \tag{24}$$

In conclusion, we find

$$\Pr[E_{\text{decr}}^*] = \Pr\left[|a_d| \geq \varsigma\right], \quad \varsigma := \sqrt{\frac{\lambda_1 + \tilde{\zeta}}{\lambda_1 + |\lambda_d|}}. \tag{25}$$

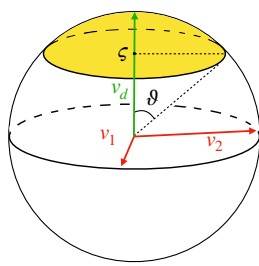

Figure 8: Illustration for the proof of Lemma 3. Any vector on the unit sphere whose angle that is less than $\vartheta = \cos^{-1}(\varsigma)$ away from $\mathbf{v}_d$ belongs to cap colored in yellow. Our goal is to bound the surface area of this spherical cap.

Therefore, since $\mathbf{a} = (a_1, a_2, \ldots, a_d)$ is uniformly distributed on the surface of the unit sphere in $\mathbb{R}^d$, we have reduced the problem to finding the surface of a spherical cap (see Figure 8). From (13), we directly get

$$\left[\frac{1}{2}(1 - \varsigma^2)\right]^{\frac{d}{2}} \leq \Pr[E^*_{\text{decr}}] \leq 2\sqrt{d-2}\left[1 - \varsigma^2\right]^{\frac{d}{2}-1}. \tag{26}$$

Plugging in $\zeta = \frac{1}{2}|\lambda_d|\sigma_2^2$, we get $1 - \zeta^2 = \frac{|\lambda_d|}{2(\lambda_1 + |\lambda_d|)}$, so by setting $\gamma := |\lambda_d|$ and $L_1 := \max\{\lambda_1, |\lambda_d|\}$ (cf. definition SOSP in Equation 1):

$$\frac{\gamma}{4L_1} \leq \frac{1}{2}(1 - \zeta^2) \leq \frac{1}{4}, \tag{27}$$

this completes the proof. $\qquad\square$

### A.3 Analysis for general function

**Lemma 2.** *Let $f(\cdot)$ be $L_1$-smooth and $\|\nabla f(\mathbf{x}_k)\| \geq \epsilon$. Algorithm 1 with $\sigma_1 = \epsilon/(L_1\sqrt{2\pi d})$ yields $\mathbf{E}[f(\mathbf{x}_{k+1}) - f(\mathbf{x}_k)|\mathbf{x}_k] \leq -\Omega\left(\frac{\epsilon^2}{L_1 d}\right)$, where $\mathbf{E}[\cdot|\mathbf{x}_k]$ denotes the conditional expectation w.r.t. $\mathbf{x}_k$.*

*Proof.* One can show (see e.g. Lemma 3.4 in [6]) that $\mathbf{E}_{\mathbf{s}_1 \sim \mathcal{S}^{d-1}}[\nabla f(\mathbf{x}_k)^\top \mathbf{s}_1|\mathbf{x}_k] = \frac{1}{\sqrt{\mu_d}}\|\nabla f(\mathbf{x}_k)\|$, with $\mu_d := 2\pi d$. Using smoothness, we obtain

$$\mathbf{E}[f(\mathbf{x}_{k+1}) - f(\mathbf{x}_k)|\mathbf{x}_k] \tag{28}$$

$$\leq \mathbf{E}[\nabla f(\mathbf{x}_k)^\top(\mathbf{x}_{k+1} - \mathbf{x}_k)|\mathbf{x}_k] + \frac{L_1}{2}\mathbf{E}[\|\mathbf{x}_{k+1} - \mathbf{x}_k\|^2] \tag{29}$$

$$\leq -\frac{\sigma_1}{\sqrt{\mu_d}}\|\nabla f(\mathbf{x}_k)\| + \frac{L_1}{2}\sigma_1^2 \tag{30}$$

$$\leq -\frac{\sigma_1}{\sqrt{\mu_d}}\epsilon + \frac{L_1}{2}\sigma_1^2, \tag{31}$$

where in the first inequality we used the fact that we can choose between $\mathbf{s}_1$ and $-\mathbf{s}_1$, and update with the perturbation which yields the best (i.e. the negative) step. Plugging-in our choice for $\sigma_1$ (which optimizes the quadratic upper bound above) we get the result. $\qquad\square$

**Lemma 4.** *Let $f(\cdot)$ be $L_1$-smooth and $L_2$-Hessian-Lipschitz. Assume $\|\nabla f(\mathbf{x}_k)\| \leq \epsilon$ and $\lambda_{\min}(\nabla^2 f(\mathbf{x}_k)) \leq -\gamma = -\epsilon^{2/3}$. Then Algorithm 1 with $\sigma_2 = \frac{\epsilon^{2/3}}{2L_2}$ is s.t.*

$$\mathbf{E}[f(\mathbf{x}_{k+1}) - f(\mathbf{x}_k)|\mathbf{x}_k] \leq -\Omega\left(\left(\frac{\gamma}{4L_1}\right)^{\frac{d}{2}}\epsilon^2\right). \tag{4}$$

*Proof.* Since $f(\mathbf{x})$ is $L_2$-Lipschitz Hessian, we have (see e.g. [39])

$$f(\mathbf{x}_{k+1}) - f(\mathbf{x}_k) \tag{32}$$

$$\leq (\mathbf{x}_{k+1} - \mathbf{x}_k)^\top \nabla f(\mathbf{x}_k) + \frac{1}{2}(\mathbf{x}_{k+1} - \mathbf{x}_k)^\top \nabla^2 f(\mathbf{x}_k)(\mathbf{x}_{k+1} - \mathbf{x}_k) + \frac{L_2}{6}\|\mathbf{x}_{k+1} - \mathbf{x}_k\|^3. \tag{33}$$

We use Lemma 3 on the quadratic $\tilde{f}(\mathbf{x}) := \frac{1}{2}(\mathbf{x} - \mathbf{x}_k)^\top \nabla^2 f(\mathbf{x}_k)(\mathbf{x} - \mathbf{x}_k)$ to guarantee a decrease of $\gamma \sigma_2^2/2$ with probability $p_{\text{decr}} = \left(\frac{\gamma}{4L_1}\right)^{d/2}$. Therefore, with probability $p_{\text{decr}}$,

$$f(\mathbf{x}_{k+1}) - f(\mathbf{x}_k) \tag{34}$$

$$\leq (\mathbf{x}_{k+1} - \mathbf{x}_k)^\top \nabla f(\mathbf{x}_k) + \frac{1}{2}(\mathbf{x}_{k+1} - \mathbf{x}_k)^\top \nabla^2 f(\mathbf{x}_k)(\mathbf{x}_{k+1} - \mathbf{x}_k) + \frac{L_2}{6}\|\mathbf{x}_{k+1} - \mathbf{x}_k\|^3 \tag{35}$$

$$\leq -\gamma \sigma_2^2/2 + \frac{L_2}{6}\sigma_2^3, \tag{36}$$

where we can ensure that $(\mathbf{x}_{k+1} - \mathbf{x}_k)^\top \nabla f(\mathbf{x}_k) \leq 0$ by testing for both $\mathbf{s}_2$ and $-\mathbf{s}_2$ in Algorithm 2 — which does not affect $\frac{1}{2}\mathbf{s}_2^\top \nabla^2 f(\mathbf{x}_k)\mathbf{s}_2$.

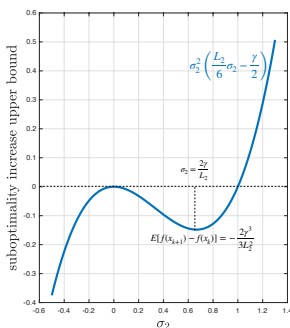

Figure 9: Selection of the value of $\sigma_2$ which yields the best decrease.

Next, we seek to minimize Equation 36 with respect to $\sigma_2$. To this, we take the derivative and set it to zero: $\sigma_2 = 0$ is a local maximizer, while $\sigma_2 = 2\gamma/L_2$ is the unique local minimizer for $\sigma_2 \geq 0$. Hence, since Equation 36 goes to infinity for $\sigma_2 \to \infty$, this minimizer is global (see Figure 9). For this value of $\sigma_2$, we have

$$f(\mathbf{x}_{k+1}) - f(\mathbf{x}_k) \leq -\frac{2}{3}\frac{\gamma^3}{L_2^2}. \tag{37}$$

Therefore, for $\gamma = \epsilon^{2/3}$, we have $f(\mathbf{x}_{k+1}) - f(\mathbf{x}_k) \leq \Omega(\epsilon^2)$ for $\mathbf{s}_2 \in E^*_{\text{decr}}$, defined in Equation 20. We proceed by computing the expected decrease using the law of total expectation

$$\mathbf{E}[f(\mathbf{x}_{k+1}) - f(\mathbf{x}_k)] \tag{38}$$

$$= \mathbf{E}[f(\mathbf{x}_{k+1}) - f(\mathbf{x}_k)|E^*_{\text{decr}}] \cdot \Pr[E^*_{\text{decr}}] + \mathbf{E}[f(\mathbf{x}_{k+1}) - f(\mathbf{x}_k)|(E^*_{\text{decr}})^c] \cdot \Pr[(E^*_{\text{decr}})^c] \tag{39}$$

$$\leq \mathbf{E}[f(\mathbf{x}_{k+1}) - f(\mathbf{x}_k)|E^*_{\text{decr}}] \cdot \Pr[E^*_{\text{decr}}] \tag{40}$$

$$= -p_{\text{decr}} \cdot \Omega(\epsilon^2). \tag{41}$$

where in the first inequality we used the fact that, by the algorithm definition, $f(\mathbf{x}_{k+1}) - f(\mathbf{x}_k) = 0$ (rejected step). $\qquad\square$

# B   Analysis Random Search PI (Algorithm 2)

**Lemma 7.** *Let $f(\cdot)$ be $L_1$-smooth and $L_2$-Hessian-Lipschitz, and assume $\|\nabla f(\mathbf{x}_k)\| \leq \epsilon$ and $\lambda_{\min}(\nabla^2 f(\mathbf{x}_k)) \leq -\gamma = -\epsilon^{2/3}$. Under Assumption 2, RSPI (Algorithm 2) with $\sigma_2 = \frac{\gamma}{2L_2}$ (choice as Theorem 5) yields $\mathbf{E}[f(\mathbf{x}_{k+1}) - f(\mathbf{x}_k)|\mathbf{x}_k] \leq -\Omega(\epsilon^2)$, independent of the problem dimension.*

*Proof.* Since $f(\mathbf{x})$ is $L_2$-Lipschitz Hessian, under Assumption 2 we have

$$\mathbf{E}[f(\mathbf{x}_{k+1}) - f(\mathbf{x}_k)|\mathbf{x}_k] \tag{42}$$

$$\leq \mathbf{E}[(\mathbf{x}_{k+1} - \mathbf{x}_k)^\top \nabla f(\mathbf{x}_k)|\mathbf{x}_k] + \frac{1}{2}\mathbf{E}\left[(\mathbf{x}_{k+1} - \mathbf{x}_k)^\top \nabla^2 f(\mathbf{x}_k)(\mathbf{x}_{k+1} - \mathbf{x}_k)|\mathbf{x}_k\right] \tag{43}$$

$$+ \frac{L_2}{6}\|\mathbf{x}_{k+1} - \mathbf{x}_k\|^3 \tag{44}$$

$$\leq -\gamma\sigma_2^2/2 + \gamma\sigma_2^2/4 + \frac{L_2}{6}\sigma_2^3 \tag{45}$$

$$= \sigma_2^2\left(-\frac{1}{4}\gamma + \frac{L_2}{6}\sigma_2\right), \tag{46}$$

where we can ensure that $(\mathbf{x}_{k+1} - \mathbf{x}_k)^\top \nabla f(\mathbf{x}_k) \leq 0$ by testing for both $\mathbf{s}_2$ and $-\mathbf{s}_2$ in Algorithm 2 (which does not affect $\frac{1}{2}\mathbf{s}_2^\top \nabla^2 f(\mathbf{x}_k)\mathbf{s}_2 + \frac{L_2}{6}\|\mathbf{s}_2\|^3$).

We therefore require $\sigma_2 \leq \frac{3}{2L_2}\gamma$ for the RHS in Eq. (36) to be negative. Choosing, as for the random search case, $\sigma_2 = \frac{\gamma}{2L_2}$,

$$\mathbf{E}[f(\mathbf{x}_{k+1}) - f(\mathbf{x}_k)|\mathbf{x}_k] \leq -\frac{1}{24}\frac{\gamma^3}{L_2^2}. \tag{47}$$

For $\gamma = \epsilon^{2/3}$, we obtain $\mathbf{E}[f(\mathbf{x}_{k+1}) - f(\mathbf{x}_k)|\mathbf{x}_k] \leq -\Omega(\epsilon^2)$. $\qquad\square$

# C   Analysis of DFPI (Algorithm 3)

## C.1   Proof of Lemma 6

We show that line 7 in Algorithm 3 can be written as a noisy power iteration step. That is,

$$\mathbf{s}_2^{(t+1)} = \mathbf{s}_2^{(t)} - \eta\frac{\mathbf{g}_+ - \mathbf{g}_-}{2r} \stackrel{\text{to show}}{=} (\mathbf{I} - \eta\nabla^2 f(\mathbf{x}))\mathbf{s}_2^{(t)} + \boldsymbol{\xi}_{\text{DFPI}}^{(t)},$$

where $\boldsymbol{\xi}_{\text{DFPI}}^{(t)}$ is an approximation error. We show that $\boldsymbol{\xi}_{\text{DFPI}}^{(t)}$ can be made as small as needed if finite difference hyperparameters $r, c$ are chosen small enough. Therefore, Alg. 3 can be seen as a noisy power method; hence one can motivate the rate in Lemma 9 using the results in [37, 29, 5], with an argument similar to [10] (remark after the Assumption 1 of this reference).

**Lemma 6.** *Let $f(\cdot)$ be $L_1$-smooth and $L_2$-Hessian-Lipschitz. The iteration of DFPI can be seen as a step of a noisy power method: $\mathbf{s}_2^{(t+1)} = (\mathbf{I} - \eta\nabla^2 f(\mathbf{x}))\mathbf{s}_2^{(t)} + \boldsymbol{\xi}_{DFPI}^{(t)}$, with $\|\boldsymbol{\xi}_{DFPI}^{(t)}\| = \mathcal{O}(rL_2 + \frac{c}{r}L_1\sqrt{d})$. In particular, $\|\boldsymbol{\xi}_{DFPI}^{(t)}\| \to 0$ as $r, \frac{c}{r} \to 0$; hence the error can be made as small as needed within the limits of numerical stability. In addition, if $f(\cdot)$ is quadratic, we have $\|\boldsymbol{\xi}_{DFPI}^{(t)}\| = 0$.*

*Proof.* We note that $\mathbf{g}_+$ and $\mathbf{g}_-$ are the finite-difference approximations of $\nabla f(\mathbf{x} + r\mathbf{s}_2^{(t)})$ and $\nabla f(\mathbf{x} - r\mathbf{s}_2^{(t)})$, respectively:

$$\mathbf{g}_+ = \sum_{i=1}^{d} \frac{f(\mathbf{x} + r\mathbf{s}_2^{(t)} + c\mathbf{e}_i) - f(\mathbf{x} + r\mathbf{s}_2^{(t)} - c\mathbf{e}_i)}{2c}\mathbf{e}_i, \tag{48}$$

$$\mathbf{g}_- = \sum_{i=1}^{d} \frac{f(\mathbf{x} - r\mathbf{s}_2^{(t)} + c\mathbf{e}_i) - f(\mathbf{x} - r\mathbf{s}_2^{(t)} - c\mathbf{e}_i)}{2c}\mathbf{e}_i. \tag{49}$$

where $r, c > 0$. Let us denote by $\boldsymbol{\xi}_{1,+}^{(t)}$ and $\boldsymbol{\xi}_{1,-}^{(t)}$ the approximation errors in the estimation of $\nabla f(\mathbf{x} + r\mathbf{s}_2^{(t)})$ and $\nabla f(\mathbf{x} - r\mathbf{s}_2^{(t)})$, respectively (properties of this error discussed at the end of the proof). We have:

$$\frac{\mathbf{g}_+ - \mathbf{g}_-}{2r} = \frac{\nabla f(\mathbf{x} + r\mathbf{s}_2^{(t)}) + \boldsymbol{\xi}_{1,+}^{(t)} - \nabla f(\mathbf{x} - r\mathbf{s}_2^{(t)}) - \boldsymbol{\xi}_{1,-}^{(t)}}{2r} \tag{50}$$

$$= \nabla^2 f(x)\mathbf{s}_2^{(t)} + \boldsymbol{\xi}_2^{(t)} + \frac{\boldsymbol{\xi}_{1,+}^{(t)} - \boldsymbol{\xi}_{1,-}^{(t)}}{2r} \tag{51}$$

$$= \nabla^2 f(x)\mathbf{s}_2^{(t)} + \boldsymbol{\xi}_{\mathrm{DFPI}}^{(t)}, \tag{52}$$

where $\boldsymbol{\xi}_2^{(t)}$ is the error on the Hessian-vector product. To conclude the proof, we bound the errors $\boldsymbol{\xi}_2^{(t)}$ and $\boldsymbol{\xi}_{1,\pm}^{(t)}$.

**Bound on $\boldsymbol{\xi}_2^{(t)}$.** This error vanishes as $r \to 0$ under Assumption 1 (see main paper):

$$\|\boldsymbol{\xi}_2^{(t)}\| = \left\| \frac{\nabla f(\mathbf{x} + r\mathbf{s}_2^{(t)}) - \nabla f(\mathbf{x} - r\mathbf{s}_2^{(t)})}{2r} - \nabla^2 f(\mathbf{x})\mathbf{s}_2^{(t)} \right\| \tag{53}$$

$$= \left\| \frac{\int_0^1 \nabla^2 f(\mathbf{x} - r\mathbf{s}_2^{(t)} + 2ur\mathbf{s}_2^{(t)})2r\mathbf{s}_2^{(t)} du}{2r} - \nabla^2 f(\mathbf{x})\mathbf{s}_2^{(t)} \right\| \tag{54}$$

$$\leq \int_0^1 \left\| \nabla^2 f(\mathbf{x} + (2u-1)r\mathbf{s}_2^{(t)}) - \nabla^2 f(\mathbf{x}) \right\| du \tag{55}$$

$$\leq rL_2 \int_0^1 |2u-1| du \tag{56}$$

$$= \frac{rL_2}{2}, \tag{57}$$

where the second equality follows directly from the fundamental theorem of calculus (see e.g. the introductory chapter in [47], proof of Lemma 1.2.2). The first inequality comes from Cauchy–Schwarz and the definition of operator norm, after noting that $\|\mathbf{s}_2^{(t)}\| = 1$. The second inequality from Hessian Lipschitzness. Note that for quadratics $L_2 = 0$ so $\xi^{(t)}_2$ is identically zero.

**Bound on $\boldsymbol{\xi}_{1,\pm}^{(t)}$.** These error also vanish as $c \to 0$, and the proof is similar to the one above. This was already shown e.g. in Lemma 3 (Appendix D) from [31]. We give a proof for completeness, again based on the fundamental theorem of calculus.

$$\mathbf{g}_+ = \frac{1}{2c} \sum_{i=1}^d \left( f(\mathbf{x} + r\mathbf{s}_2^{(t)} + c\mathbf{e}_i) - f(\mathbf{x} + r\mathbf{s}_2^{(t)} - c\mathbf{e}_i) \right) \mathbf{e}_i \tag{58}$$

$$= \sum_{i=1}^d \mathbf{e}_i \mathbf{e}_i^\top \int_0^1 \nabla f(\mathbf{x} + r\mathbf{s}_2^{(t)} + (2u-1)c\mathbf{e}_i) du. \tag{59}$$

Therefore, using the subadditivity of the Euclidean norm and gradient Lipschitzness,

$$\|\boldsymbol{\xi}_{1,+}^{(t)}\|^2 = \left\| \sum_{i=1}^d \mathbf{e}_i \mathbf{e}_i^\top \int_0^1 \left( \nabla f(\mathbf{x} + r\mathbf{s}_2^{(t)} + (2u-1)c\mathbf{e}_i) - \nabla f(\mathbf{x} + r\mathbf{s}_2^{(t)}) \right) du \right\|^2 \tag{60}$$

$$\leq \sum_{i=1}^d \left( \int_0^1 \left\| \nabla f(\mathbf{x} + r\mathbf{s}_2^{(t)} + (2u-1)c\mathbf{e}_i) - \nabla f(\mathbf{x} + r\mathbf{s}_2^{(t)}) \right\| du \right)^2 \tag{61}$$

$$\leq \sum_{i=1}^d L^2 c^2 \left( \int_0^1 |2u-1| \right)^2 \tag{62}$$

$$= \frac{dL^2 c^2}{4}, \tag{63}$$

where the first inequality holds true because the vectors in the sum are mutually orthogonal and $\|\mathbf{e}_i \mathbf{e}_i^\top\|^2 = 1$. Note that here an additional dependency on the dimension comes in — which is due to the triangle inequality and the nature of the estimator (sum of $d$ terms). The same bound can be derived for $\boldsymbol{\xi}_{1,-}^{(t)}$. This concludes the proof.

**The quadratic case.** As mentioned above, in the quadratic case the Hessian is constant; hence $L_2 = 0$ and therefore $\|\boldsymbol{\xi}_2^{(t)}\| = 0$. However, from the bound above it seems that the bound on $\|\boldsymbol{\xi}_{1,\pm}^{(t)}\|$ does not vanish, since $L_1 \neq 0$. This is an artefact of the proof technique. Indeed, for the quadratic case we have $\mathbf{g}_+ = f(\mathbf{x} + r\mathbf{s}_2^{(t)})$ and $\mathbf{g}_- = f(\mathbf{x} - r\mathbf{s}_2^{(t)})$. This can be seen by inspecting the integral in Equation 59: assuming $f(\mathbf{x}) = C + (\mathbf{x} - \mathbf{x}^*)^\top \mathbf{H}(\mathbf{x} - \mathbf{x}^*)$ we have

$$\int_0^1 \nabla f(\mathbf{x} + r\mathbf{s}_2^{(t)} + (2u-1)c\mathbf{e}_i)du = \int_0^1 \mathbf{H}(\mathbf{x} + r\mathbf{s}_2^{(t)} + (2u-1)c\mathbf{e}_i - \mathbf{x}^*)du \tag{64}$$

$$= \mathbf{H}(\mathbf{x} + r\mathbf{s}_2^{(t)} - \mathbf{x}^*) + \mathbf{H}\int_0^1 (2u-1)c\mathbf{e}_i du \tag{65}$$

$$= \mathbf{H}(\mathbf{x} + r\mathbf{s}_2^{(t)} - \mathbf{x}^*) \tag{66}$$

$$= \nabla f(\mathbf{x} + r\mathbf{s}_2^{(t)}). \tag{67}$$

This concludes the proof. $\qquad\square$

## C.2 Lemma 9 and results on convergence of (noisy) power methods

Finding the smallest eigenvalue (assumed to be negative) of the Hessian $\nabla^2 f(\mathbf{x}_t)$) is equivalent to the one of finding the largest eigenvalue of $\mathbf{A} = \mathbf{I} - \eta\nabla^2 f(\mathbf{x}_t)$, where $\eta$ is a small positive number such that $\eta \leq 1/\|\nabla^2 f(\mathbf{x}_t)\|$ (the spectral norm of $\nabla^2 f(\mathbf{x}_t)$). For this choice of $\eta$, $\mathbf{I} - \eta\nabla^2 f(\mathbf{x}_t)$ is positive semidefinite, hence one can use an (inexact) power method to retrieve the maximum eigenvalue. We first present the standard error analysis of the power iteration (which we adapt from [25]), assuming we have access to the true Hessian. Then, we discuss the setting where we can only compute approximate Hessian-vector products (analysis adapted from [29]). Finally, we present the bound for the Derivative-Free Power Iteration (DFPI) algorithm (Alg. 3).

### C.2.1 Warm-up: error analysis for the exact power method

Let $\mathbf{A} \in \mathbb{R}^{d \times d}$ be a positive definite matrix with eigenvalues $a_1 > a_2 \geq \ldots a_d > 0$, and corresponding eigenvectors $\mathbf{v}_1, \mathbf{v}_2, \ldots, \mathbf{v}_d$. Eigenvalues are counted together with their algebraic multiplicity. We seek an approximation for the dominant eigendirection $\mathbf{v}_1$. The power method on the positive semidefinite matrix $\mathbf{A}$ can be found as Algorithm 4.

---

**Algorithm 4** POWER METHOD (EXACT, ACCESS TO HESSIAN-VECTOR PRODUCTS REQUIRED)

1: **INPUT :** A matrix $\mathbf{A} \in \mathbb{R}^{d \times d}$ with eigenvalues $a_1 > a_2 \geq \ldots a_d > 0$.
2: Randomly initialize $\mathbf{v}^{(0)} \sim \mathcal{S}^{d-1}$
3: **for** $t = 0 \ldots T-1$ **do**
4: $\quad \mathbf{v}^+ = \mathbf{A}\mathbf{v}^{(t)}$
5: $\quad \mathbf{v}^{(t+1)} = \mathbf{v}^+/\|\mathbf{v}^+\|$
6: **end for**
7: **OUTPUT :** $\mathbf{v}^{(T)}$ approximating $\mathbf{v}_1$, leading eigenvector of $\mathbf{A}$.

---

We present the fundamental yet simple result, confirming that the power iteration step decreases the distance to the dominant eigendirection. We recall that $\angle(\mathbf{v}, \mathbf{u}) := \arccos \frac{\langle \mathbf{v}, \mathbf{u} \rangle}{\|\mathbf{u}\| \cdot \|\mathbf{v}\|}$.

**Lemma 14.** *Consider a step of Alg. 4),* $\tan(\angle(\mathbf{v}^{(t+1)}, \mathbf{v}_1)) \leq \frac{a_2}{a_1} \tan(\angle(\mathbf{v}^{(t)}, \mathbf{v}_1))$.

*Proof.* First, we write $\mathbf{v}^{(t)}$ in the eigenbasis $\{\mathbf{v}_i\}_{i=1}^d$: $\mathbf{v}^{(t)} = \sum_{i=1}^d \alpha_i^{(t)}\mathbf{v}_i$. Crucially, note that

$$\tan(\angle(\mathbf{v}^{(t)}, \mathbf{v}_1)) = \frac{\sqrt{\sum_{j=2}^d (\alpha_j^{(t)})^2}}{\alpha_1^{(t)}}. \tag{68}$$

Since $\mathbf{v}^+ = \sum_{i=1}^d a_i\alpha_i^{(t)}\mathbf{v}_i$, we have that

$$\tan(\angle(\mathbf{v}^{(t+1)}, \mathbf{v}_1)) = \tan(\angle(\mathbf{v}^+, \mathbf{v}_1)) = \frac{\sqrt{\sum_{j=2}^d a_j^2 (\alpha_j^{(t)})^2}}{a_1 \alpha_1^{(t)}} \leq \frac{a_2}{a_1} \tan(\angle(\mathbf{v}^{(t)}, \mathbf{v}_1)). \quad (69)$$

$\square$

As noted by [29], the dependence on the eigenvalue separation arises already in the classical perturbation argument of Davis-Kahan [17]. If $a_1$ has multiplicity greater than 1, then of course the ratio will be $a_k/a_1$, where $a_k$ is the first eigenvalue strictly smaller than $a_1$. More on this point can be found in Remark 2.

From the lemma above, we can easily deduce the error on the eigenvalue computation

**Theorem 15.** *Algorithm 4 outputs a vector $\mathbf{v}^{(T)}$ such that $|(\mathbf{v}^{(T)})^\top \mathbf{A} \mathbf{v}^{(T)} - a_1| \leq \epsilon a_1$ if*

$$T \geq \frac{a_1}{2(a_1 - a_2)} \log\left(\frac{\tan^2(\angle(\mathbf{v}^{(0)}, \mathbf{v}_1))}{\epsilon}\right). \quad (70)$$

*Moreover, as also mentioned in Lemma 2.5 in [29] and Lemma 2.2 in [5], if $\mathbf{v}^{(0)}$ is randomly initialized on the unit sphere, the main result in [50] implies that with probability $1 - \delta - e^{\Omega(d)}$ we have $\tan^2(\angle(\mathbf{v}^{(0)}, \mathbf{v}_1)) \leq d/\delta^2$. Hence, with probability $1 - \delta - e^{\Omega(d)}$, we have*

$$T \geq \frac{a_1}{2(a_1 - a_2)} \log\left(\frac{d}{\epsilon \delta^2}\right). \quad (71)$$

*Proof.* Note that since $\mathbf{v}^{(T)}$ is normalized,

$$\tan^2(\angle(\mathbf{v}^{(T)}, \mathbf{v}_1))^2 = \frac{\sum_{j=2}^d (\alpha_j^{(T)})^2}{(\alpha_1^{(T)})^2} = \frac{1 - (\alpha_1^{(T)})^2}{(\alpha_1^{(T)})^2}, \quad (72)$$

therefore

$$(\alpha_1^{(T)})^2 = \frac{1}{1 + \tan^2(\angle(\mathbf{v}^{(T)}, \mathbf{v}_1))}, \qquad \sum_{j=2}^d (\alpha_j^{(T)})^2 = \frac{\tan^2(\angle(\mathbf{v}^{(T)}, \mathbf{v}_1))^2}{1 + \tan^2(\angle(\mathbf{v}^{(T)}, \mathbf{v}_1))}. \quad (73)$$

We have the following bound:

$$|(\mathbf{v}^{(T)})^\top \mathbf{A} \mathbf{v}^{(T)} - a_1| = \left| a_1 (\alpha_1^{(T)})^2 + \sum_{i=2}^d a_i (\alpha_i^{(T)})^2 - a_1 \right| \quad (74)$$

$$= a_1 - a_1 (\alpha_1^{(T)})^2 - \sum_{i=2}^d a_i (\alpha_i^{(T)})^2 \quad (75)$$

$$\leq a_1 - a_1 \frac{1}{1 + \tan^2(\angle(\mathbf{v}^{(T)}, \mathbf{v}_1))} - a_d \frac{\tan^2(\angle(\mathbf{v}^{(T)}, \mathbf{v}_1))}{1 + \tan^2(\angle(\mathbf{v}^{(T)}, \mathbf{v}_1))} \quad (76)$$

$$= \frac{\tan^2(\angle(\mathbf{v}^{(T)}, \mathbf{v}_1))}{1 + \tan^2(\angle(\mathbf{v}^{(T)}, \mathbf{v}_1))} (a_1 - a_d) \quad (77)$$

$$\leq a_1 \tan^2(\angle(\mathbf{v}^{(T)}, \mathbf{v}_1)) \quad (78)$$

where the second equality is given by the fact that $a_1$ is the biggest eigenvalue of $\mathbf{A}$. All in all, we need $\tan^2(\angle(\mathbf{v}^{(T)}, \mathbf{v}_1))$ to be smaller than $\epsilon/(a_1 - a_d)$. Thanks to Lemma 14, we have that

$$\tan^2(\angle(\mathbf{v}^{(T)}, \mathbf{v}_1)) \leq \left(\frac{a_2}{a_1}\right)^{2T} \tan^2(\angle(\mathbf{v}^{(0)}, \mathbf{v}_1)). \quad (79)$$

Therefore, we require $\left(\frac{a_2}{a_1}\right)^{2T} a_1 \tan^2(\angle(\mathbf{v}^{(0)}, \mathbf{v}_1)) \leq \epsilon a_1$, which can be written as,

$$\left(\frac{a_1}{a_2}\right)^{2T} \geq \frac{\tan^2(\angle(\mathbf{v}^{(0)}, \mathbf{v}_1))}{\epsilon}. \quad (80)$$

We conclude by taking the $\log$ on both sides:

$$T \geq \frac{1}{2\log(a_1/a_2)} \log\left(\frac{\tan^2(\angle(\mathbf{v}^{(0)}, \mathbf{v}_1))}{\epsilon}\right) \tag{81}$$

Since, for all $x \in \mathbb{R}$, $\log(x) \geq 1 - \frac{1}{x}$ and $a_1 > a_2$, the above expression is verified if

$$T \geq \frac{a_1}{2(a_1 - a_2)} \log\left(\frac{\tan^2(\angle(\mathbf{v}^{(0)}, \mathbf{v}_1))}{\epsilon}\right) \tag{82}$$

$\square$

**Remark 1.** *Note that Theorem 15 is exactly equivalent to Theorem 8.2.1 in [25]. Here we followed a proof more similar to the one in [29].*

**Remark 2** (Eigen-gap dependency)**.** *The bound in Theorem 15 depends on the eigen-gap $a_1 - a_2$: as $a_1$ and $a_2$ get closer, the result suggests that we need a very large number of iterations to find a good approximation of $\mathbf{v}_1$. This is true because the power method is confounded by $\mathbf{v}_2$, and takes a long time to "decide" which one between $\mathbf{v}_1$ and $\mathbf{v}_2$ is dominant. However, this of course does not imply that the complexity in finding $\mathbf{v}^{(T)}$ such that $|(\mathbf{v}^{(T)})^\top \mathbf{A}\mathbf{v}^{(T)} - a_1| \leq \epsilon$ increases — this is an artefact of our simple analysis (inspired by [29, 25, 5]), which crucially goes through Lemma 14 to derive the bound. Indeed, as the next theorem shows, **it is possible to directly remove this dependency**.*

**Theorem 16** (Consequence of Thm. 3.1 and Thm. 4.1 in [37])**.** *Let $\mathbf{v}^{(0)}$ be initialized randomly on the surface of the unit sphere. The power method returns a vector $\mathbf{v}^{(T)}$ such that $|(\mathbf{v}^{(T)})^\top \mathbf{A}\mathbf{v}^{(T)} - a_1| \leq \epsilon a_1$ in $T = \mathcal{O}(\log(d)/\epsilon)$ iterations, in expectation. For the result to hold with probability $1 - \delta$, one instead needs at least $T = \mathcal{O}(\log(d/\delta^2)/\epsilon)$ iterations.*

This results in also cited in [56], where the bound above is used to conclude that, if $\lambda_{\min}(\nabla^2 f(\mathbf{x})) \leq -\gamma$ and $\|\nabla^2 f(\mathbf{x})\| \leq L_1$, the power method on $(\mathbf{I} - \eta\nabla^2 f(\mathbf{x}))$ finds a direction $\mathbf{v}^{(T)}$ such that, with probability $1 - \delta$, $(\mathbf{v}^{(T)})^\top \nabla^2 f(\mathbf{x})\mathbf{v}^{(T)} \leq -\frac{\gamma}{2}$ in $\mathcal{O}\left(\frac{L_1}{\gamma} \log(d/\delta^2)\right)$ iterations.

This proves directly a version of Lemma 9 for the case of vanishing error.

**Lemma 17** (Noiseless version of Lemma 9)**.** *Let the parameters of DFPI be such that the error $\boldsymbol{\xi}_{DFPI}$ is vanishing (possible within the limits of numerical stability by Lemma 6). Let $\eta \leq 1/L_1$. Let $\gamma = \epsilon^{2/3}$; for a fixed RSPI iteration, $T_{DFPI} = \mathcal{O}\left(\epsilon^{-2/3} L_1 \log\left(\frac{d}{\delta^2}\right)\right)$ DFPI iterations are enough to ensure validity of Assumption 2 (without the expectation sign) at $\mathbf{x}_k$ with probability $1 - \delta - e^{\Omega(d)}$.*

*Proof.* Direct consequence of the reasoning above, supported by Lemma 6. $\square$

### C.2.2 Error analysis for the noisy power method

We now consider the case where $\mathbf{A}\mathbf{v}$ cannot be computed exactly (Algorithm 5): we denote by $\boldsymbol{\xi}^{(t)}$ the error in computing the Hessian-vector product $\mathbf{A}\mathbf{v}^{(t)}$.

---

**Algorithm 5** POWER METHOD (NOISY, APPROXIMATE HESSIAN-VECTOR PRODUCTS PERMITTED)

1: **INPUT :** A matrix $\mathbf{A}$ with eigenvalues $a_1 > a_2 \geq \ldots a_d$.
2: Randomly initialize $\mathbf{v}^{(0)} \sim \mathcal{S}^{d-1}$
3: **for** $t = 0 \ldots T - 1$ **do**
4:     $\mathbf{v}^+ = \text{approx}(\mathbf{A}\mathbf{v}^{(t)}) = \mathbf{A}\mathbf{v}^{(t)} + \boldsymbol{\xi}^{(t)}$
5:     $\mathbf{v}^{(t+1)} = \mathbf{v}^+/\|\mathbf{v}^+\|$
6: **end for**
7: **OUTPUT :** $\mathbf{v}^{(T)}$, approximating $\mathbf{v}_1$, leading eigenvector of $\mathbf{A}$

---

We are now ready to state the main result we are going to use on the noisy power method, presented in the main text in a less precise way, as Lemma 9. This result was first derived in [29], and can be seen as an extension to Theorem 15. In plain english: **for small enough noise**, the bound in Theorem 15 still holds with arbitrarily high probability.

**Theorem 18** (Direct consequence of Corollary 1.1 in [29]). *In the context of Algorithm 5, fix the desired accuracy $\epsilon \le 1/2$ and a failure probability $\delta$. Assume that for all iterations $t$ the noise is small enough: (1) $5\|\boldsymbol{\xi}^{(t)}\| \le \epsilon(a_1 - a_2)$ and (2) $5|\mathbf{v}_1^\top \boldsymbol{\xi}^{(t)}| \le \delta(a_1 - a_2)/\sqrt{d}$. With probability $1 - \delta - e^{-\Omega(d)}$, Algorithm 5 returns $\mathbf{v}^{(T)}$ such that $|(\mathbf{v}^{(T)})^\top \mathbf{A}\mathbf{v}^{(T)} - a_1| \le \epsilon a_1$ if*

$$T \ge \mathcal{O}\left( \frac{a_1}{a_1 - a_2} \log\left( \frac{d}{\epsilon\delta^2} \right) \right). \tag{83}$$

*Proof.* In the proof of Theorem 15, we showed that

$$|(\mathbf{v}^{(T)})^\top \mathbf{A}\mathbf{v}^{(T)} - a_1| \le \tan^2(\angle(\mathbf{v}^{(T)}, \mathbf{v}_1)) \cdot (a_1 - a_d). \tag{84}$$

This is enough to complete the result given Corollary 1.1 in [29]. □

The proof of Lemma 9 then follows from a generalization of Theorem 16 to the noisy case (under the requirement of small enough noise). This is possible since the bounds in Theorem 18 and Theorem 15 are equivalent — meaning that the geometry of convergence is not drastically affected by noise.

### C.3 How to speed up DFPI with SPSA: an experimental motivation

We study some interesting properties of the SPSA gradient estimator, introduced by [53], in the context of DFPI (Algorithm 3, main paper). In particolar, we consider using SPSA instead of finite-difference(FD), which is the base for our theory (Thm. 10)

$$\mathbf{g}_\pm^{\text{SPSA}} = \sum_{i=1}^d \frac{f(\mathbf{x} \pm r\mathbf{s}_2^{(t)} + c\boldsymbol{\Delta}) - f(\mathbf{x} \pm r\mathbf{s}_2^{(t)} - c\boldsymbol{\Delta})}{2c\Delta_i} \mathbf{e}_i.$$

$\rightarrow$ **4 function evaluations** to get estimates of $\nabla f(\mathbf{x} \pm \mathbf{s}_2^{(t)})$.

$$\mathbf{g}_+^{\text{FD}} = \sum_{i=1}^d \frac{f(\mathbf{x} \pm r\mathbf{s}_2^{(t)} + c\mathbf{e}_i) - f(\mathbf{x} \pm r\mathbf{s}_2^{(t)} - c\mathbf{e}_i)}{2c} \mathbf{e}_i$$

$\rightarrow$ **4$d$ function evaluations** to get estimates of $\nabla f(\mathbf{x} \pm \mathbf{s}_2^{(t)})$.

**The SPSA estimator is asymptotically unbiased, but variance might be independent of the hyperparamerter $c$.** Consider $f(x_1, x_2) = x_1^2 - x_2^2$, we want to approximate its gradient using SPSA. Since perturbation is $(\Delta_1, \Delta_2)$, we have $f(\mathbf{x} + c\boldsymbol{\Delta}) - f(\mathbf{x} - c\boldsymbol{\Delta}) = 4c\Delta_1 x_1 - \Delta_2 x_2$. Therefore $\mathbf{g}^{\text{SPSA}} = \sum_{i=1}^2 \frac{f(\mathbf{x}+c\boldsymbol{\Delta})-f(\mathbf{x}-c\boldsymbol{\Delta})}{2c\Delta_i} \mathbf{e}_i = \sum_{i=1}^2 \frac{2\Delta_1 x_1 - 2\Delta_2 x_2}{\Delta_i} \mathbf{e}_i$. Since $\Delta_i$ are Bernoulli, then $\mathbf{E}[\mathbf{g}^{\text{SPSA}}] = \nabla f$. However, the estimator variance is finite and independent of $c$.

**Experimental comparison.** From the result in the paragraph above, one might conclude that SPSA cannot provide a satisfactory approximation of Hessian-vector products, and therefore cannot be used as a valid alternative to FD in the context of an approximate power method such as DFPI. However, in Figure 10 & 11 we show that, for small enough $\eta$, the update $\mathbf{s}_2^{(t+1)} = \mathbf{s}_2^{(t)} - \eta \frac{\mathbf{g}_+^{\text{SPSA}} - \mathbf{g}_-^{\text{SPSA}}}{2r}$, $\mathbf{s}_2^{(t+1)} = \mathbf{s}_2^{(t+1)}/\|\mathbf{s}_2^{(t+1)}\|$ can effectively build a vector $\mathbf{s}_2$ aligned with negative curvature, even as the problem dimension increases. In these experiments, we consider applying DFPI to estimate the negative curvature direction $\mathbf{e}_d$ of $f(\mathbf{x}) = \mathbf{x}^\top diag(\lambda_1, \lambda_2, \cdots, \lambda_d)\mathbf{x}$, with $\lambda_d < 0$ (non-axis aligned case discussed later). As we saw in Prop. 8, the finite difference estimator yields an exact power method on this function. Instead, SPSA yields an inexact power method where the error is independent of $r, c$ (see last paragraph and Fig. 12). As expected, SPSA does not actually converge to the leading eigenvector. However, it can always be tuned to yield an approximation which satisfies Assumption 2, in a total number of function evaluations which is actually smaller than FD. Further research is needed to better understand this phenomenon. However, this motivates the use of SPSA as a cheap alternative to FD in DFPI. In the experiments section of the main paper, we indeed show that this approximation is enough to yield a satisfactory improvement over vanilla method which do not consider computing negative curvature. As can be evinced from the last paragraph and from the proof of Prop. 8, the results in this case are independent of the values of $r$ and $c$; however, they could in principle depend on the landscape rotation. We show in Figure 12 that this is not the case using two random rotations.

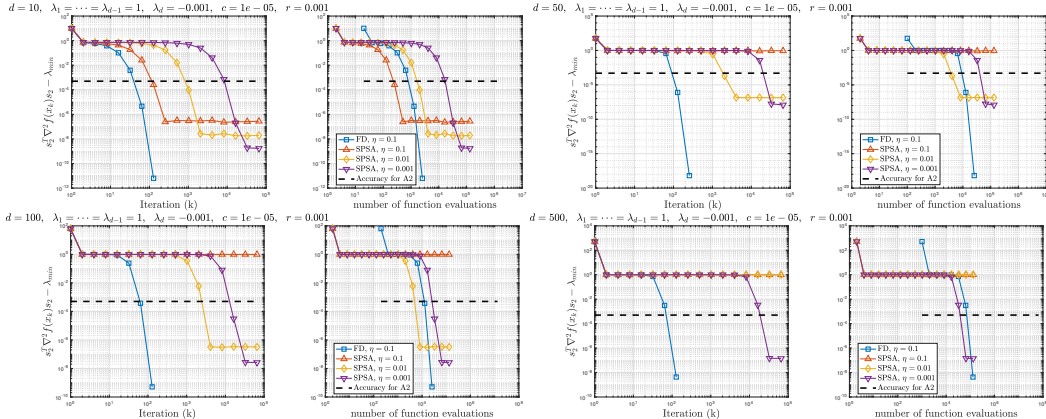

Figure 10: Experiment 1: $f(\mathbf{x}) = \mathbf{x}^\top diag(\lambda_1, \lambda_2, \cdots, \lambda_d)\mathbf{x}$, $\lambda_d = -0.001$. Settings described in the paragraph above.

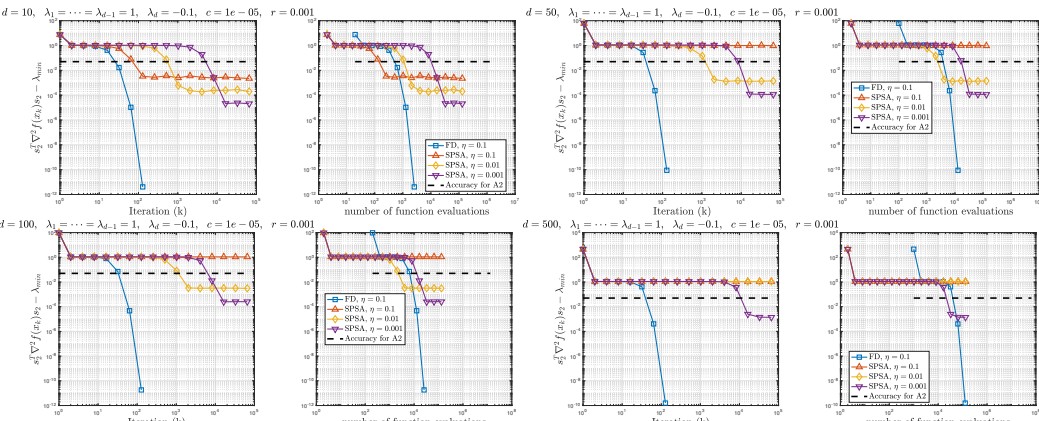

Figure 11: Experiment 2: $f(\mathbf{x}) = \mathbf{x}^\top diag(\lambda_1, \lambda_2, \cdots, \lambda_d)\mathbf{x}$, $\lambda_d = -0.1$. Settings described in the paragraph above.

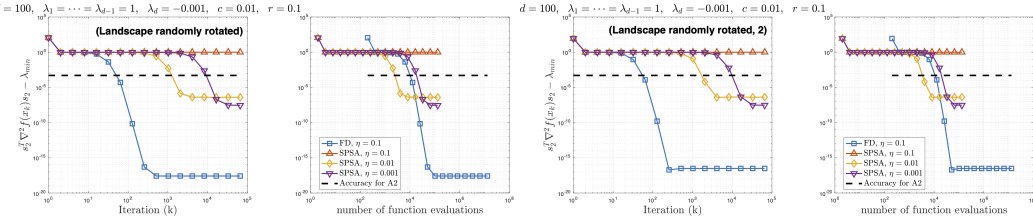

Figure 12: Experiment 2: $f(\mathbf{x}) = \mathbf{x}^\top \boldsymbol{U}^\top diag(\lambda_1, \lambda_2, \cdots, \lambda_d)\boldsymbol{U}\mathbf{x}$, $\lambda_d = -0.1$, where $\boldsymbol{U}$ is a random orthogonal matrix. Dynamics for two two different random orthogonal matrices are shown, where we additionally also decreased $c$ and $r$. The evolution is similar to the one in Figure 10, showing that SPSA is robust to both landscape rotations and hyperparameter choice.

# D   Experimental Results

All of our experiments are conducted on the Google Colaboratory [8] environment without any hardware accelerators.

## D.1   Function with growing dimension

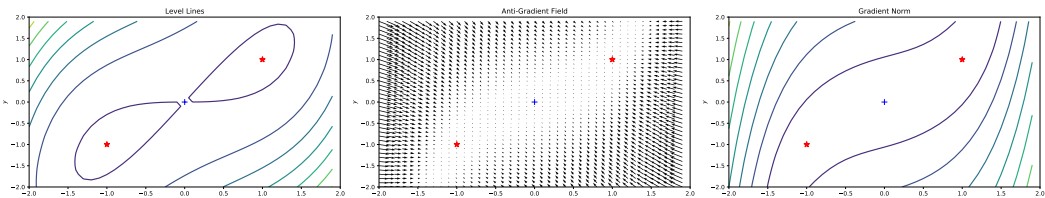

Figure 13: The landscape of the objective $f(x_1, \cdots, x_d, y) = \frac{1}{4}\sum_{i=1}^{d} x_i^4 - y\sum_{i=1}^{d} x_i + \frac{d}{2}y^2$ for $d = 1$. A blue cross denotes a strict saddle point, whereas a red star corresponds to a global minimizer.

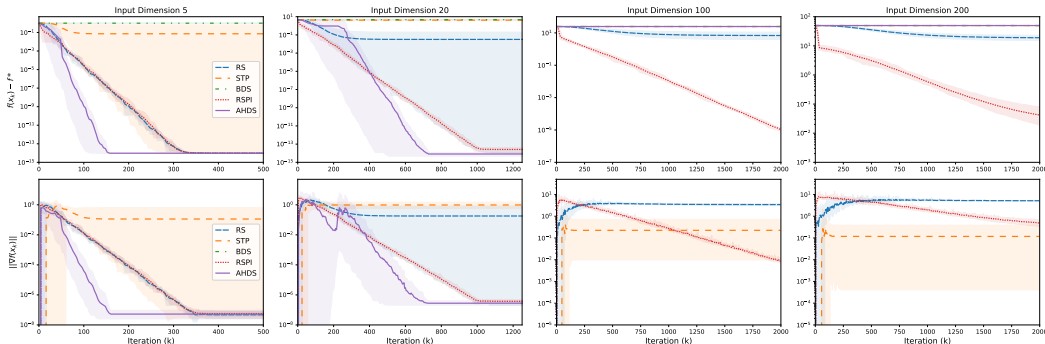

Figure 14: Empirical performance while minimizing $f(x_1, \cdots, x_d, y) = \frac{1}{4}\sum_{i=1}^{d} x_i^4 - y\sum_{i=1}^{d} x_i + \frac{d}{2}y^2$ against the number of iterations. Confidence intervals show min-max intervals over ten runs. All algorithms are initialized at the strict saddle point across all runs.

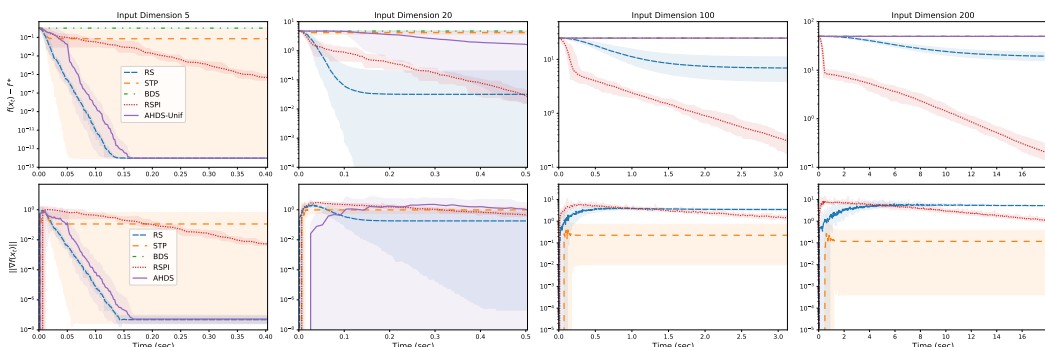

Figure 15: Empirical performance while minimizing the objective defined in the main paper against wall-clock time. Confidence intervals show min-max intervals over ten runs. All algorithms are initialized at the strict saddle point across all runs.

## D.2 Rastrigin function

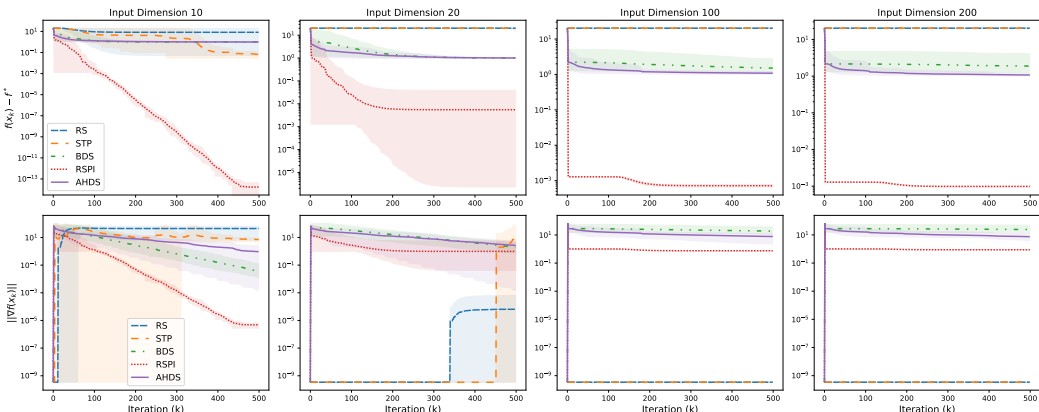

Figure 16: Empirical performance while minimizing the Rastrigin function against the number of iterations. Confidence intervals show min-max intervals over ten runs. All algorithms are initialized at a strict saddle point across all runs.

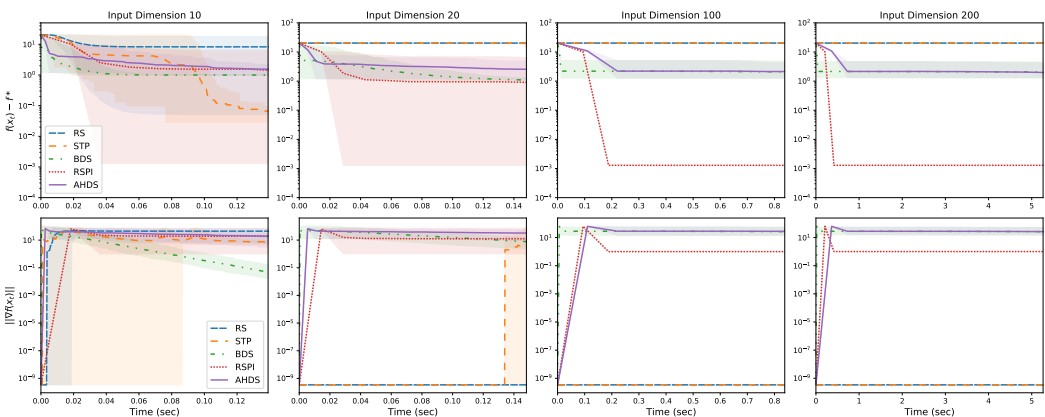

Figure 17: Empirical performance while minimizing the Rastrigin function against wall-clock time. Confidence intervals show min-max intervals over ten runs. All algorithms are initialized at a strict saddle point across all runs.

**Initialization process.** The critical points of the Rastrigin function satisfy

$$x_i + 10\pi \sin(2\pi x_i) = 0 \tag{85}$$

for all $i = 1, ..., d$. The point $\mathbf{x} = \mathbf{0}$ is the unique global minimizer. Stationary points include local minimizers, local maximizers and saddle points. One solution is given by $x_i \approx 0.503$ (truncated to three decimal points). We consider the following initialization

$$x_i = \begin{cases} 0.503 & \text{if } i \in \mathcal{I} \\ 0 & \text{otherwise,} \end{cases} \tag{86}$$

where $\mathcal{I}$ is a set of coordinates with cardinality strictly smaller than $d$. In this setup, each non-zero coordinate will be a direction of negative curvature. If we set $\mathcal{I} = \{1, ..., d\}$ the point $\mathbf{x}$ is a local maximizer.

In our experiments we choose $\mathcal{I}$ to have a single coordinate (picked randomly in each experiment repetition). Based on Lemma 3, we expect that having a single direction of negative curvature will

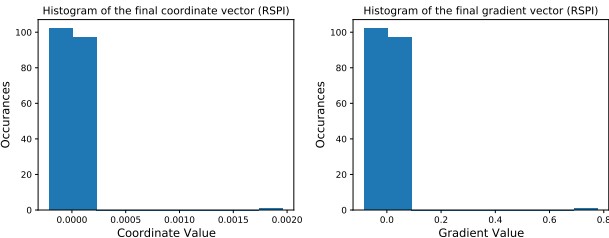

Figure 18: Histogram of the point coordinates (left) and gradient values (right) at the final iterate of RSPI for $d = 200$.

challenge the core mechanism of each algorithm while trying to escape the saddle, especially as the input dimension increases. The results in Figure 16 support our theoretical argument that as the problem dimension grows the probability of sampling a direction that is aligned with the direction of negative curvature decreases exponentially. As a result, RS and STP fail to escape the saddle point for $d = 100, 200$.

The rest of the algorithms converge quickly (for $d = 100, 200$ there is no significant progress after 25 iterations). We speculate that this behaviour is related to the initialization choice. Figure 18 shows the distribution of the point coordinates and gradient values at the final iterate of RSPI for $d = 200$. In both plots, we observe a cluster of values around zero and a stand-alone component. The later corresponds to the same coordinate that was initialized to non-zero in order to give rise to a saddle point. We observe that the coordinate moves closer to zero (the final coordinate value is less than $0.0020$, whereas the initial value was $0.503$) where the global minimizer occurs. This improvement is achieved through the successful usage of DFPI. That is, RSPI successfully approximates the direction of negative curvature in order to escape the saddle point and move closer to the minimum. Afterwards, no significant progress is achieved via random sampling and that is why the performance curve flattens out after a few iterations. The reason is that in order to achieve further progress via random sampling, it is required to sample a direction that aligns with the single direction of non-zero gradient (see Figure 18 (right)) and we expect that probability to decrease exponentially as the dimension increases. That is why further progress can be achieved for $d = 10, 20$ but not for $d = 100, 200$.

### D.3 Leading eigenvector problem

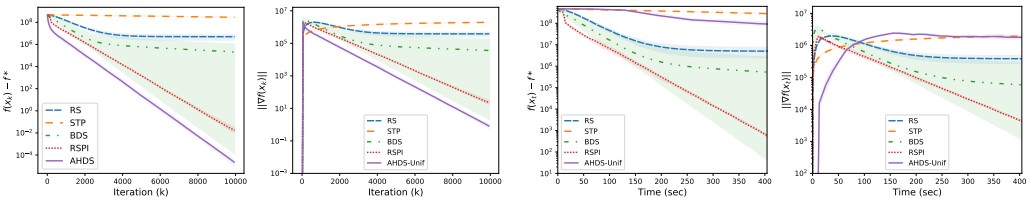

Figure 19: Empirical performance in finding the leading eigenvector of a 350-dimensional random matrix against iteration and wall-clock time. Confidence intervals show min-max intervals over ten runs. All algorithms are initialized at a strict saddle point across all runs.

# E  Algorithm Descriptions

---

**Algorithm 6** Stochastic Three Points (STP)

---
1: **INPUTS :** $\eta_0 \in \mathbb{R}^+, \phi : \mathbb{R}^+ \to \mathbb{R}^+$
2: Initialize $\mathbf{x}_0$
3: **for** $k = 0 \ldots K$ **do**
4:     $\mathbf{s}_k \sim \mathcal{S}^{d-1}$
5:     $\mathbf{x}_{k+1} = \arg\min \left\{ f(\mathbf{x}_k), f(\mathbf{x}_k + \eta_k \mathbf{s}_k), f(\mathbf{x}_k - \eta_k \mathbf{s}_k) \right\}$
6:     $\eta_{k+1} = \phi(\eta_0)$
7: **end for**

---

**Algorithm 7** Basic Direct Search (BDS)

---
1: **INPUTS :** $\eta_{\max} > \eta_0 > 0, \gamma > 1 > \theta > 0, \rho : \mathbb{R}^+ \to \mathbb{R}^+$
2: Set $k = 0$ and initialize $\mathbf{x}_0$.
3: Generate a polling set $\mathcal{D}_k$.
4: If there exists $\mathbf{s}_k \in \mathcal{D}_k$ such that

$$f(\mathbf{x}_k + \eta_k \mathbf{s}_k) < f(\mathbf{x}_k) - \rho(\eta_k)$$

then declare the iteration **successful**, set $\mathbf{x}_{k+1} = \mathbf{x}_k + \eta_k \mathbf{s}_k$, $\eta_{k+1} = \min\{\gamma \eta_k, \eta_{\max}\}$, $k = k+1$ and go to step 4.
5: Otherwise, declare the iteration **unsuccessful**, set $\mathbf{x}_{k+1} = \mathbf{x}_k$, $\eta_{k+1} = \theta \eta_k$, $k = k + 1$ and go to step 4.

---

**Algorithm 8** Approximate Hessian Direct Search (AHDS)

---
1: **INPUTS :** $\eta_{\max} > \eta_0 > 0, \gamma > 1 > \theta > 0, \rho : \mathbb{R}^+ \to \mathbb{R}^+$
2: Set $k = 0$ and initialize $\mathbf{x}_0$.
3: Generate a polling set $\mathcal{D}_k$. If there exists $\mathbf{s} \in \mathcal{D}_k$ such that

$$f(\mathbf{x}_k + \eta_k \mathbf{s}) < f(\mathbf{x}_k) - \rho(\eta_k) \tag{87}$$

then declare iteration $k$ successful with $\mathbf{s}_k = \mathbf{s}$ and go to 8. Otherwise go to 5.
4: If there exists $\mathbf{s} \in \mathcal{D}_k$ such that Eq. (87) is satisfied with $-\mathbf{s}$, then declare the iteration successful with $\mathbf{s}_k = -\mathbf{s}$ and go to step 8. Otherwise, go to step 6.
5: Choose $\mathcal{B}_k$ as a subset of $\mathcal{D}_k$ with $d$ linearly independent directions, which we index by $\mathbf{u}_1, ..., \mathbf{u}_d$. If there exists $\mathbf{s} \in \{\mathbf{u}_i + \mathbf{u}_j, 1 \le i < j \le d\}$ such that Eq. (87) holds, then declare the iteration successful with $\mathbf{s}_k = \mathbf{s}$ and go to step 8. Otherwise, go to step 7.
6: Define the Hessian approximation at iteration $k$ as

$$(H_k)_{i,j} = \frac{f(\mathbf{x}_k + \eta_k \mathbf{u}_i) - f(\mathbf{x}_k) + f(\mathbf{x}_k - \eta_k \mathbf{u}_i)}{\eta_k^2} \quad \text{if } i = j,$$

and

$$(H_k)_{i,j} = \frac{f(\mathbf{x}_k + \eta_k \mathbf{u}_i + \eta_k \mathbf{u}_j) - f(\mathbf{x}_k + \eta_k \mathbf{u}_i) - f(\mathbf{x}_k + \eta_k \mathbf{u}_j) + f(\mathbf{x}_k)}{\eta_k^2} \quad \text{if } i < j,$$

for all $i, j \in \{1, ..., d\}^2$. Compute a unitary eigenvector $\mathbf{v}_k$ associated with the minimum eigenvalue of $H_k$. If $\mathbf{v}_k$ or $-\mathbf{v}_k$ satisfy the decrease condition in Eq. (87), then declare the iteration successful with $\mathbf{s}_k$ equal to $\mathbf{v}_k$ or $-\mathbf{v}_k$. Otherwise, declare the iteration unsuccessful and go to step 8.
7: If the iteration was successful, set $\mathbf{x}_{k+1} = \mathbf{x}_k + \eta_k \mathbf{s}_k$ and $\eta_{k+1} = \min\{\gamma \eta_k, \eta_{\max}\}$. Otherwise, set $\mathbf{x}_{k+1} = \mathbf{x}_k$ and $\eta_{k+1} = \theta \eta_k$.
8: Increment $k$ and go to step 4.

---

# F Hyperparameter selection

For all tasks, the hyperparameters of each method are selected based on a coarse grid search procedure that is refined heuristically by trial and error. The hyperparameters of RS and RSPI are initialized and updated in the same manner, hence the only difference between the two is that RSPI explicitly extracts negative curvature whereas the two-step RS samples a direction at random. In our experiments, we keep $\sigma_2$ constant and only update $\sigma_1$ every $T_{\sigma_1} \in \mathbb{Z}^+$ iterations using the update rule $\sigma_1 \leftarrow \rho\sigma_1$ where $\rho \in (0, 1)$. The parameters $\rho$ and $T_{\sigma_1}$ are also selected based on a coarse grid search. We run DFPI for 20 iterations for all the results shown in the paper and we clarify in the following tables whether Finite Differences (DFPI-FD) or SPSA (DFPI-SPSA) is used to approximate the gradient evaluations within DFPI.

We illustrate the effect that some crucial parameters have on the performance of the two-step Random Search and the Random Search PI algorithms. In the following figures, confidence intervals show min-max intervals across five runs. All algorithms are initialized at the strict saddle point of the objective

$$f(x_1, \cdots, x_d, y) = \frac{1}{4}\sum_{i=1}^{d} x_i^4 - y\sum_{i=1}^{d} x_i + \frac{d}{2}y^2. \tag{88}$$

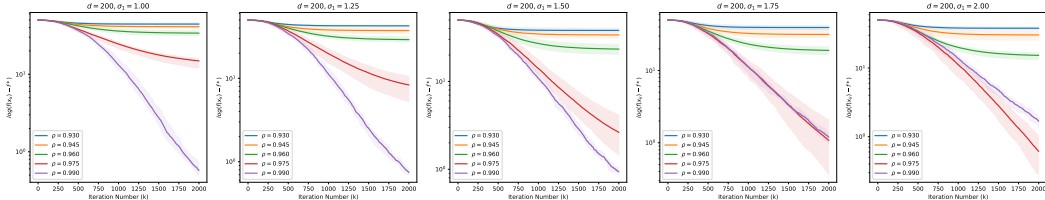

Figure 20: Empirical behaviour of the vanilla RS algorithm while minimizing the objective defined in Eq. 88 for $d = 200$ across different settings of the pair of parameters $(\sigma_1, \rho)$. The parameter $T_{\sigma_1}$ is fixed to 10.

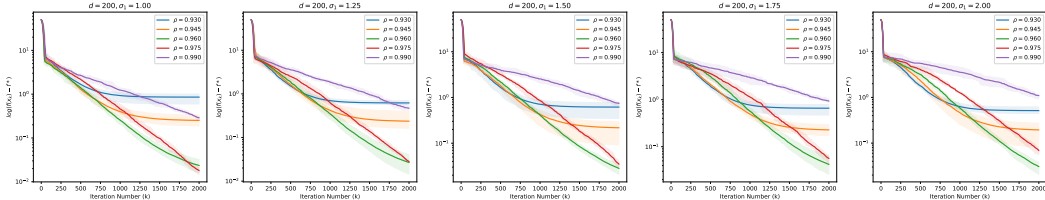

Figure 21: Empirical behaviour of RSPI while minimizing the objective defined in Eq. 88 for $d = 200$ across different settings of the pair of parameters $(\sigma_1, \rho)$. The parameter $T_{\sigma_1}$ is fixed to 10.

Table 1: Hyperparameters for the leading eigenvector task.

| Method | Parameters |
|---|---|
| **d = 350** | |
| RS | $\sigma_1 = 9.25, \sigma_2 = 4.5, \rho = 0.97, T_{\sigma_1} = 25, T_{\text{DFPI}} = 20$ |
| RSPI | $\sigma_1 = 9.25, \sigma_2 = 4.5, \rho = 0.97, T_{\sigma_1} = 25, T_{\text{DFPI}} = 20$, DFPI-SPSA |
| BDS | $\eta_0 = 5.8, \eta_{max} = 35.0, \gamma = 1.25, \theta = 0.5, \rho(x) = 0$ |
| AHDS | $\eta_0 = 5.8, \eta_{max} = 35.0, \gamma = 1.25, \theta = 0.5, \rho(x) = 0$ |

Table 2: Hyperparameters for the objective in Eq. (88).

| Method | Parameters |
|---|---|
| **d = 5** | |
| RS | $\sigma_1 = 1.8, \sigma_2 = 0.65, \rho = 0.6, T_{\sigma_1} = 10, T_{\text{DFPI}} = 20$ |
| RSPI | $\sigma_1 = 1.8, \sigma_2 = 0.65, \rho = 0.6, T_{\sigma_1} = 10, T_{\text{DFPI}} = 20$, DFPI-SPSA |
| STP | $\eta_0 = 2.5, \phi(\eta_k) = 0.5\eta_k$ if $k \equiv \mod 10$ (every 10 iterations) |
| BDS | $\eta_0 = 0.8, \eta_{max} = 10.0, \gamma = 1.25, \theta = 0.5, \rho(x) = 0$ |
| AHDS | $\eta_0 = 0.8, \eta_{max} = 10.0, \gamma = 1.25, \theta = 0.5, \rho(x) = 0$ |
| **d = 20** | |
| RS | $\sigma_1 = 1.75, \sigma_2 = 0.65, \rho = 0.78, T_{\sigma_1} = 15, T_{\text{DFPI}} = 20$ |
| RSPI | $\sigma_1 = 1.75, \sigma_2 = 0.65, \rho = 0.78, T_{\sigma_1} = 15, T_{\text{DFPI}} = 20$, DFPI-SPSA |
| STP | $\eta_0 = 2.5, \phi(\eta_k) = 0.5\eta_k$ if $k \equiv \mod 10$ (every 10 iterations) |
| BDS | $\eta_0 = 0.8, \eta_{max} = 10.0, \gamma = 1.25, \theta = 0.5, \rho(x) = 0$ |
| AHDS | $\eta_0 = 0.8, \eta_{max} = 10.0, \gamma = 1.25, \theta = 0.5, \rho(x) = 0$ |
| **d = 100** | |
| RS | $\sigma_1 = 1.0, \sigma_2 = 0.65, \rho = 0.95, T_{\sigma_1} = 15, T_{\text{DFPI}} = 20$ |
| RSPI | $\sigma_1 = 1.0, \sigma_2 = 0.65, \rho = 0.95, T_{\sigma_1} = 15, T_{\text{DFPI}} = 20$, DFPI-SPSA |
| STP | $\eta_0 = 2.5, \phi(\eta_k) = 0.5\eta_k$ if $k \equiv \mod 10$ (every 10 iterations) |
| BDS | $\eta_0 = 5.0, \eta_{max} = 20.0, \gamma = 1.25, \theta = 0.5, \rho(x) = 0$ |
| AHDS | $\eta_0 = 5.0, \eta_{max} = 20.0, \gamma = 1.25, \theta = 0.5, \rho(x) = 0$ |
| **d = 200** | |
| RS | $\sigma_1 = 1.75, \sigma_2 = 0.65, \rho = 0.96, T_{\sigma_1} = 15, T_{\text{DFPI}} = 20$ |
| RSPI | $\sigma_1 = 1.75, \sigma_2 = 0.65, \rho = 0.96, T_{\sigma_1} = 15, T_{\text{DFPI}} = 20$, DFPI-SPSA |
| STP | $\eta_0 = 2.5, \phi(\eta_k) = 0.5\eta_k$ if $k \equiv \mod 10$ (every 10 iterations) |
| BDS | $\eta_0 = 5.0, \eta_{max} = 20.0, \gamma = 1.25, \theta = 0.5, \rho(x) = 0$ |
| AHDS | $\eta_0 = 5.0, \eta_{max} = 20.0, \gamma = 1.25, \theta = 0.5, \rho(x) = 0$ |

Table 3: Hyperparameters for the Rastrigin function.

| Method | Parameters |
|---|---|
| **d = 10** | |
| RS | $\sigma_1 = 0.25, \sigma_2 = 0.25, \rho = 0.83, T_{\sigma_1} = 5, T_{\text{DFPI}} = 20$ |
| RSPI | $\sigma_1 = 0.25, \sigma_2 = 0.25, \rho = 0.83, T_{\sigma_1} = 5, T_{\text{DFPI}} = 20, \text{DFPI-FD}$ |
| STP | $\eta_0 = 0.25, \phi(\eta_k) = \eta_0/\sqrt{k+1}$ |
| BDS | $\eta_0 = 0.25, \eta_{max} = 10.0, \gamma = 1.1, \theta = 0.9, \rho(x) = 0$ |
| AHDS | $\eta_0 = 0.25, \eta_{max} = 10.0, \gamma = 1.1, \theta = 0.9, \rho(x) = 0$ |
| **d = 20** | |
| RS | $\sigma_1 = 0.255, \sigma_2 = 0.25, \rho = 0.83, T_{\sigma_1} = 5, T_{\text{DFPI}} = 20$ |
| RSPI | $\sigma_1 = 0.255, \sigma_2 = 0.25, \rho = 0.83, T_{\sigma_1} = 5, T_{\text{DFPI}} = 20, \text{DFPI-FD}$ |
| STP | $\eta_0 = 0.25, \phi(\eta_k) = \eta_0/\sqrt{k+1}$ |
| BDS | $\eta_0 = 0.25, \eta_{max} = 10.0, \gamma = 1.1, \theta = 0.9, \rho(x) = 0$ |
| AHDS | $\eta_0 = 0.25, \eta_{max} = 10.0, \gamma = 1.1, \theta = 0.9, \rho(x) = 0$ |
| **d = 100** | |
| RS | $\sigma_1 = 0.15, \sigma_2 = 0.25, \rho = 0.83, T_{\sigma_1} = 5, T_{\text{DFPI}} = 20$ |
| RSPI | $\sigma_1 = 0.15, \sigma_2 = 0.25, \rho = 0.83, T_{\sigma_1} = 5, T_{\text{DFPI}} = 20, \text{DFPI-FD}$ |
| STP | $\eta_0 = 0.25, \phi(\eta_k) = \eta_0/\sqrt{k+1}$ |
| BDS | $\eta_0 = 0.25, \eta_{max} = 10.0, \gamma = 1.1, \theta = 0.9, \rho(x) = 0$ |
| AHDS | $\eta_0 = 0.25, \eta_{max} = 10.0, \gamma = 1.1, \theta = 0.9, \rho(x) = 0$ |
| **d = 200** | |
| RS | $\sigma_1 = 0.15, \sigma_2 = 0.25, \rho = 0.83, T_{\sigma_1} = 5, T_{\text{DFPI}} = 20$ |
| RSPI | $\sigma_1 = 0.15, \sigma_2 = 0.25, \rho = 0.83, T_{\sigma_1} = 5, T_{\text{DFPI}} = 20, \text{DFPI-FD}$ |
| STP | $\eta_0 = 0.25, \phi(\eta_k) = \eta_0/\sqrt{k+1}$ |
| BDS | $\eta_0 = 0.25, \eta_{max} = 10.0, \gamma = 1.1, \theta = 0.9, \rho(x) = 0$ |
| AHDS | $\eta_0 = 0.25, \eta_{max} = 10.0, \gamma = 1.1, \theta = 0.9, \rho(x) = 0$ |