# OpenReview forum: "On the Second-order Convergence Properties of Random Search Methods"
_NeurIPS.cc/2021/Conference — NeurIPS 2021 Poster_

### Official Review · Reviewer_huRV · 2021-07-06

**Rating:** 7
**Confidence:** 3

**Summary:**

This paper proposed a novel random search method that can converge to a second-order stationary point at a faster rate for high-dimensional problems with only zeroth-order oracle under standard Lipschitz gradient Lipschitz Hessian assumptions. The number of function evaluations is only linear in the problem dimension. Experiments verify the theoretical results.

**Limitations And Societal Impact:**

Yes.

**Main Review:**

Originality: This paper proposed a new random search method to solve high-dimensional problems. This paper is well written and technically adequate. Noisy power iteration based on only zeroth-order oracle (DFPI) is used to estimate the negative eigenvector and avoid the storage of Hessian. It is shown that if the estimation is good enough, the number of outer iteration needed in direct search or the number of calculations of the negative eigenvector is independent of dimension. DFPI provably converges with iteration complexity that is nearly linearly dependent on dimensional, therefore the final computation complexity avoids exponential dependency. I would like to see more zero-order methods for high-dimensional problems are compared in experiments and analyzed in theory.

Quality: First, theoretical results show the proposed method can achieve better performance on the worst-case analysis than the previous random search methods for high-dimensional problems. Algorithm 2 is similar to the framework of the naive random search method. Algorithm 3 adopts a zeroth-order strategy. Is there any equivalence or intuition between the proposed method and zeroth-order methods? Besides, how about the comparison of the iteration complexity between the proposed method and the AHDS method for high-dimensional problems?

Second, experimental results show the proposed method can achieve better performance than the previous random search methods for high-dimensional problems. However, zeroth-order methods are another popular type of derivative-free method. My concern is whether the proposed method can achieve better performance than zeroth-order methods for high-dimensional non-convex problems in experiments.

Some other questions:
1. Is this paper the first to introduce STP with two different stepsizes? If the authors could comment on this point it might be helpful in appreciating the novelty here.
2. Which one of DFPI-FD and DFPI-SPSA is used for each experiment and why? On line 265, it says "We choose to run DFPI with SPSA for 20 iterations for all the results shown in the paper." but in Table 3, it shows DFPI-FD.
3. Figure 10-12 gives a comparison of FD and SPSA in terms of spectrum error. However, is there any comparison on convergence speed of RSPI with DFPI-FD and DFPI-SPSA?
4. In conclusion, it says "the dependency in terms of the function input dimension becomes linear, a result which we clearly observed in our experimental results". What do experiments ensure us that the dependency is linear?
5. In Lemma 9 and Theorem 15, why the probability is $1- \delta-e^{\Omega(d)}$ instead of $1- \delta^{\Omega(1)}-e^{-\Omega(d)}$?

Clarity: The paper is clearly written and well-organized.

Significance: The results are significant to the relevant problems, but may not have a broad impact.

**Time Spent Reviewing:**

8

---

> ### Author Response · Authors · 2021-08-09
> **Answering questions**
>
> We thank the reviewer for their feedback.
>
> ### Equivalence/Intuition related to zeroth-order methods
> We are not aware of any zero-order method similar to our RSPI algorithm. However, we think it would be possible to design such a method taking inspiration from Carmon et al. (2018), and Liu and Yang (2017). For a detailed discussion of the fundamental conceptual differences between direct search methods (e.g. RSPI) and methods that rely completely on gradient approximations, we invite the reader to check the STP paper (https://arxiv.org/pdf/1902.03591.pdf, page 2).
>
> ### Comparison of the iteration complexity with AHDS
> - Complexity AHDS: they first require the computation of the Hessian, and then require computing the minimum eigenvector (see AHDS paper or Algorithm 8 in appendix). Instead, our approach directly computes the most negative direction, therefore avoiding the O(d^2) computation and storing of the Hessian and the further Hessian-vector products. Our experimental section shows this speed-up very clearly
>
> - performance compared to zeroth-order methods for high-dimensional non-convex problems
> We thank the reviewer for the interesting question. To the best of our knowledge, the only modern zero-order method with similar convergence guarantees on non-convex problems is the NeurIPS paper from Flokas et al. https://arxiv.org/pdf/1910.13021.pdf . We note that this method is conceptually very different from ours, since it relies on stochastic perturbations to escape saddle points (see algorithm 2 of their paper, line 1). Indeed, their method is basically a zero-order equivalent of perturbed gradient descent – that is, it is not necessarily a descent method. Instead, to escape saddle points we rely only on the geometry of the objective function – and our algorithm is a descent method. While it would be interesting to compare zero-order methods and random search methods,we feel that a thorough comparison should also include momentum and adaptive stepsizes, e.g. complex algorithms like ZO-AdaMM (https://arxiv.org/abs/1910.06513 ). For this reason, since our contribution is of a theoretical nature, we kept the experimental setup simple and restrict the methods class to random search. This allows us to discuss the properties of our method and provide a clean comparison with similar methods.
> To sum it up, we agree that a thorough experimental comparison of zero-order vs random search methods is of interest and we will add a comment about this in the conclusion section.
>
> ### Is this paper the first to introduce STP with two different stepsizes?
> Yes, please see the explanation at the beginning of section 3.1. We use the two-step variant as a surrogate for vanilla random searches such as STP that only use one step. We could also have written the algorithm with a single adaptive step but we think the two-step approach is easier to understand. Also note that this variant is strictly better than STP in theory, but we do not necessarily see it as an interesting practical approach as it still suffers from an exponential dependency to the dimension. This is why we propose our new variant.
>
> ### Which one of DFPI-FD and DFPI-SPSA is used for each experiment and why?
> Tables 1, 2 and 3 in Appendix F clarify which one of DFPI-FD and DFPI-SPSA is used for each experiment. In each experiment, we first attempt to use DFPI-SPSA to approximate the gradient evaluations within DFPI since it is less expensive than DFPI-FD (see “How to further speed up DFPI with SPSA in Section 3 and Appendix C.3). As demonstrated by the experimental results in Section 4, in the case of the function with growing dimension and the leading eigenvector problem, the performance of RSPI was competitive while using DFPI-SPSA. On the other hand, in the case of the Rastrigin function, the performance of RSPI was found to be very poor while using DFPI-SPSA as the algorithm could not escape the saddle point across a wide range of hyperparameter values (results are not included in the paper). Instead, when DFPI-FD was used the performance of RSPI was improved significantly (for the same set of hyperparameters). The results shown in the paper for the Rastrigin function are based on DFPI-SPSA.
>
> ### "We choose to run DFPI with SPSA for 20 iterations for all the results shown in the paper."
> This comment reflects the fact that whenever DFPI-SPSA is used, 20 iterations are performed. We agree that the comment can be misleading. Thank you for pointing this out, we will address it accordingly.
>
> ### Figure 10-12 gives a comparison of FD and SPSA in terms of spectrum error
> We have only conducted experiments for RSPI with both DFPI-FD and DFPI-SPSA on the Rastrigin function. In that case, we found that RSPI with DFPI-SPSA could not escape the strict saddle point (following the initialization process described in Appendix D.2) across a wide range of hyperparameter values. Instead, when DFPI-FD was used the performance of RSPI was improved significantly as illustrated in Section 4 and Appendix D.2 (all of the results included in the paper for the Rastrigin function are based on RSPI with DFPI-FD). In the case of the function with growing dimension and the leading eigenvector problem we have only conducted experiments using RSPI with DFPI-SPSA. Since the performance of the method was found to be very competitive compared to previous methods we did not attempt to run RSPI with DFPI-FD as that would incur an additional computational cost (see “How to further speed up DFPI with SPSA in Section 3 and Appendix C.3).
>
> ### What do experiments ensure us that the dependency is linear?
> The linear dependency can be observed empirically in Figure 1. We will provide further experimental results in the updated version.
>
> ### Comment about Lemma 9 and Theorem 15
> Yes, you are correct, thank you for pointing this out, we will fix it.

---

### Official Review · Reviewer_369q · 2021-07-15

**Rating:** 6
**Confidence:** 3

**Summary:**

In this submission, the authors analyse a variant of stochastic three points method which is a derivative free optimisation algorithm using a random search routine. They prove that the method converges to a second-order stationary point but suffers a complexity that increases exponentially with respect to the dimension of the problem. They propose a modification to the algorithm so that the complexity in terms of the number of function evaluations is only linear in the problem dimension. They provide experiments showing that the new method outperforms other random search methods in escaping saddle points for large dimensional problems.

**Limitations And Societal Impact:**

There not immediate potential negative societal impact of the work.

**Main Review:**

The authors have two main contributions in this submission. First, they analyze a variant of stochastic three points (STP) method that uses a two-step design which aims to take first and seconder order information into consideration using only the function values without having access to the derivatives. The authors prove that this algorithm converges to a second-order stationary point where the derivative is zero and the Hessian is positive definite. They also show that this method’s complexity increases exponentially with respect to the dimension of the problem.

To deal with the complexity issue, they propose a modification the algorithm using a derivative-free routine which approximates the eigenvector corresponding to the smallest negative eigenvalue of the Hessian so that they capture the negative curvature. The routine they propose is based on a noisy power method and has a lower computation cost in terms of the imput dimension than the previous methods. They show that the complexity of the new algorithm in terms of the number of function evaluations is linear in the problem dimension. They also provide experiments showing that the proposed method outperforms other random search methods STP, Basic Direct Search (BDS) and the Approximate Hessian Direct Search (AHDS) in escaping saddle points when the problem dimension is large.

The theoretical contributions for the random search methods seem to be new and interesting. The submission is well written with an easy-to-follow presentation.

The experiments are limited. The authors do not provide any performance evaluation of the proposed method in any of the machine learning applications mentioned in the introduction. Although, the new method seems to perform well in escaping saddle points in the synthetic problems, its general performance in the mentioned applications is not clear.

For a wide-range of optimization problems in machine learning, the evaluation of the entire objective function with a full pass through the dataset is either impossible or it is computationally very demanding. Therefore, many algorithms rely on noisy (stochastic) gradient information and use the data in batches. Given that the authors propose algorithms, which require full passes through the data, the algorithms will have limited applications in machine learning. As also mentioned by the authors, it is true that stochastic search algorithms can still be used in reinforcement learning, meta-learning, and so on. However, the authors have not provided any numerical experiments on any one of these applications.

I believe the authors consider their contribution as mainly theoretical  -which explains the limited experiments- but then without experiments on machine learning problems, I do not think this conference is the best fit for this work.

**Time Spent Reviewing:**

5 hours

---

> ### Author Response · Authors · 2021-08-08
> **Answering questions**
>
> We thank the reviewer for their feedback. We agree that a significant part of our contribution is theoretical, which we believe is a good fit for Neurips whose reviewer’s guidelines point out that “These contributions may be theoretical, methodological, algorithmic, ...”.
>
> We address your main comments below.
>
> ## Experiments
> Please note that we designed our experiments in order to verify the validity of the theoretical bounds derived in the paper. We have two sets of experiments:
> 1) First set of experiments is on relatively simple functions where the hyper-parameter tuning is relatively straightforward (unlike more complex problems discussed below). We note that our experimental setting follows common practices in the literature, see e.g. Flokas et al. Neurips 2019, https://arxiv.org/pdf/1910.13021.pdf (only a single toy problem). This also goes beyond the experimental settings of many derivative-free papers published at Neurips, e.g.:
> https://papers.nips.cc/paper/2020/file/6646b06b90bd13dabc11ddba01270d23-Paper.pdf
> https://papers.nips.cc/paper/2018/file/36d7534290610d9b7e9abed244dd2f28-Paper.pdf
> https://papers.nips.cc/paper/2012/file/e6d8545daa42d5ced125a4bf747b3688-Paper.pdf
>
> The same applies to saddle-point papers in the gradient-based settings, many papers purely make a theoretical contribution and were accepted at ICML, or Neurips, e.g.
> http://proceedings.mlr.press/v70/jin17a/jin17a.pdf
> https://papers.nips.cc/paper/2017/file/f79921bbae40a577928b76d2fc3edc2a-Paper.pdf
> https://proceedings.neurips.cc/paper/2019/file/3fb04953d95a94367bb133f862402bce-Paper.pdf
>
> Therefore, we disagree with the statement that our contribution is not a good fit for a conference such as Neurips.
>
> 2) The second set of experiments is on a leading eigenvector problem, which is a fundamental primitive of numerical linear algebra with numerous applications. In machine learning, such applications include spectral clustering, community detection, etc. We point out that this problem is of great interest in optimization (https://proceedings.neurips.cc/paper/2018/file/1b318124e37af6d74a03501474f44ea1-Paper.pdf ), since the corresponding landscape is non-convex (see e.g. http://proceedings.mlr.press/v130/alimisis21a/alimisis21a.pdf )
>
> In addition to (2), two machine learning applications where our approach would be practically interesting are adversarial black-box attacks and reinforcement learning. Both applications are major topics in ML and require expert knowledge for hyper-parameter tuning, which would require significantly extending the scope of the paper.
>
> ## Extension to noisy setting
> Thank you for pointing this out, we agree with you that this is an important practical point. In fact, our analysis can be relatively easily extended to work with stochastic estimates. For that, we would follow the probabilistic framework defined in Chen et al. 2015, see section 3 in https://arxiv.org/pdf/1504.04231.pdf.
> This framework requires a bound on the approximation error of the stochastic estimates. This can be easily satisfied using standard concentration bounds (e.g. Hoeffind’s inequality assuming bounded variance). As a result, one should be able to extend our theoretical analysis to the case where the algorithm uses batches, we will add a comment.
>
> We would be grateful to get further feedback after you read our replies, do they address your concerns? Thank you.

---

### Official Review · Reviewer_BTuv · 2021-07-15

**Rating:** 6
**Confidence:** 2

**Summary:**

This work studies the problem of optimizing a non-convex function in a deterministic setting, where the gradient of the function is not accessible. The authors provide analysis of convergence to a second order stationary point, where the gradient of the function is zero and the Hessian matrix is positive definite.

Assumptions: Let $f$ to be the objective function:

1. $f$ is lower bounded.
2. The gradient and Hessian of $f$ are Lipschitz
3. The smallest eigenvalue of the Hessian matrix has a uniform lower bound.

First the authors proposed a decent algorithm (Algorithm 1, page 4), which is a variant of vanilla random search method. Their analysis for this algorithm is divided into two parts: The case when the gradient is large (Lemma 2, page 4), and the case when the gradient is small but the smallest eigenvalue of Hessian is negative (Lemma 4, page 5). The latter is the case where they are close to a strict saddle point. Aggregating Lemmas 4 and 5 yields one of their main results, Theorem 5 (page 5). This theorem outlines the fact that convergence to a second order stationary point grows exponentially as a function of $d$ (dimension). To overcome this issue, they proposed Algorithms 2 and 3. The main idea of these algorithms is that instead of a random direction search (which causes an exponential dependency on $d$), they estimate negative curvature directions for escaping a saddle point. This new approach requires $4d$ additional function evaluations at each iteration. As they initially claimed, they could overcome the curse of dimensionality and obtain a linear dependency with respect to $d$. This result is marked out in Theorem 10 (page 7).

**Limitations And Societal Impact:**

The authors briefly discuss the limitations of their work. I also pointed out some, along with suggestions to improve the paper in the Main Review section.

Since the paper is quite theoretical, I except no direct social impact at this point.

**Main Review:**

Originality: They worked on a well studied problem. Their proposed Algorithms 2 and 3, can derive linear dependency on dimension which is outlined in Theorem 10 (page 10). This approach is inspired by Carmon et al. (2018), but instead of estimating the Hessian matrix they only estimate negative curvature directions.

Quality: 1. I read the proofs up to Section C. All the theoretical claims are well supported.

2. The authors provided an adequate literature review and gave quite enough intuition about the challenges and their methodology.

3. The paper is quite technical. In the appendix, I appreciate that the authors provided numerical experiments after critical lemmas to provide a clear intuition.

4. They did not give any sense of optimality of their main result in Theorem 10. It would be useful if the authors discussed this. Also, it would be helpful if the authors provided a table to summarize the upper bounds \ lower bounds in the relevant works.

5. Since both Carmon et al. (2018), and Liu and Yang (2017) deal with the situation where the gradient of the function is available, the provided comparison in Section 3.2 needs more discussion.

6. One point is not clear to me. In the proof of Lemma 3, why is it enough to derive a bound for equation (14)? And to be precise, why do you need the term $\nabla^2\tilde{f}(.)$ at point $x_0$?

Clarity: The paper is well written. Except for some vague parts which I asked about above and I pointed out here, the proofs are self-contained and clear.

These are some minor typos and suggestions, up to Section C:

1. In Algorithm 1, the lines 5 and 7. It is evident to me what you mean by that. But mathematically, it may be confusing for the readers when you write argmin on a set.
2. I think instead of reference [52] in line 52, you should cite the reference [38].
3. line 172. typo: you should add a parenthesis for the term $\lambda_\min$(...).
4. line 246. typo: Ass ----> Assumption, since throughout the note you wrote Assumption rather than Ass.
5. line 542. Equation (10), there is a term $(2/3)^{(\alpha + 1)}$, which should be $(1/2)^{(\alpha + 1)}$.
6. line 559. "In for our first result" must be a typo.


Significance: Inspired by Carmon et al. (2018) and Lin and Yang (2017), this paper obtained a nearly linear dependency on dimension ($d\log(d)$) for gradient-free non-convex optimization, which to the best of my knowledge is new in the literature.

I would like to thank the authors for their response. After the rebuttal and re-studying Section 3, I find the discussion provided about Carmon et al. (2018), and Liu and Yang (2017) complete and enough. Considering other reviews and the authors' feedback, my assessment of the paper remains unchanged.

**Time Spent Reviewing:**

8 hours

---

> ### Author Response · Authors · 2021-08-08
> **Answering questions**
>
> First of all, we would like to thank the reviewer for checking the proofs in detail. Your feedback is very valuable. We will fix the typos you pointed out and we answer your main questions below.
>
> - Optimality of their main result in Theorem 10:
> We will add a table as suggested, thank you. Regarding the optimality of the result, there are two aspects to consider: for the dependency to d, we don’t think this can be improved significantly since some existing lower bounds show a dependency to d. Regarding epsilon, we do not know and would need to derive a separate lower bound. This would be of interest.
>
> - Comment about Carmon et al. (2018), and Liu and Yang (2017):
> We said that our approach is “inspired” from these recent methods in the sense that they also use a special routine to extract a negative curvature direction. However, the two approaches you mention are gradient based, while our approach is gradient free so a comparison is not possible in the case where gradients are not accessible. We will clarify this point. We are happy to provide further details if the reviewer could elaborate on what sort of discussion they would like to see.
>
> - proof of Lemma 3 and equation (14):
> We apologize for the confusion and we think we should change the notation in the proof. We should have said that x_0 in the proof is any initial point. You can see at line 585 that we apply Lemma 3 to the current iterate x_k. We will change the notation in Equation (14), thank you for pointing this out!
>
> - Typos: thank you, we will fix them.
>
> We would be grateful to get further feedback after you read our replies, do they address your concerns? We weren't sure what discussion you wanted to see about Carmon et al. (2018), and Liu and Yang (2017), please let us know whether you misunderstood your comment. Thank you.

---

> > ### Comment · Reviewer_BTuv · 2021-08-25
> > **Acknowledging**
> >
> > I would like to thank the authors for their response. After the rebuttal and re-studying Section 3, I find the discussion provided about Carmon et al. (2018), and Liu and Yang (2017) complete and enough. Considering other reviews and the authors' feedback, my assessment of the paper remains unchanged.

---

> > > ### Author Response · Authors · 2021-08-27
> > > **Thank you for acknowledging our answer**
> > >
> > > Dear reviewer,
> > >
> > > Thank you for acknowledging our answer!

---

### Official Review · Reviewer_Tejo · 2021-07-31

**Rating:** 6
**Confidence:** 3

**Summary:**

This paper is related to random-search methods when optimizing non-convex objective functions without having access to derivatives. The authors provided convergence guarantees to a second-order stationary with only access to the function evaluations. Further, they showed some numerical results for highlighting their theoretical results.

**Limitations And Societal Impact:**

They have not.

**Main Review:**

1- The paper is mostly well-written.

2- SOSP can happen also when norm of gradient is small and the Hessian is positive-semi definite. The text in the first page needs to be modified, unless the authors are considering some other assumptions in their mind such as strict saddles (which in this case the authors should mention it in the text as well).

3- The contributions (in Page 2) are not well-described. Each contribution should be separate (the second contribution is the rest of the first one; needs be re-written in a better way)

4- In Line 67, do the Authors mean $\mathcal{O}(\epsilon^{-2})$?

5- In Algorithm 1, the authors need to clarify steps 5 and 7 more. Why one step is Gradient Step and the other is Curvature Step while both steps are very similar to each other? the authors need to mention how the second one is using the curvature information while the first one is not? further, they should not simply say that the choices of $\sigma_1$ and $\sigma_2$ are the main difference between gradient and curvature steps. More info is required.

6- In Lemma 3, it can happen that one of eigenvalues to be zero ($\lambda_i =0$ for some $i$) based on the assumption that $\lambda_d <0$. How the authors can escape the saddles when eigenvalues of Hessian are not strict? Usually, in order to escape from saddles and converge to SOSP, the assumption "strict saddles" is required. The authors need to mention how they can escape saddles with their assumptions.

7- Algorithm 2 is very expensive. Step 6 in Algorithm 2 requires $T_{DFPI} \times 4d$ functions evaluations, which is indeed unpractical for large dimensional problems.

8- Assumption 2 is very strong, how the authors can justify it?

9- The size of problems for the numerical experiments (both in main body and also in the appendix) are very small ($d \leq 200$). Why the authors do not consider larger dimensional problems?

10-  In Figure 3, why the methods AHDS and BDS do not improve at all? how the authors select the initial points? are they saddles? The authors need to mention how they select the initial points since in Appendix D2 (Figures 16 and 17) $||\nabla f||$ is very small from the very beginning, why it happens?

11- The authors need to mention how they tuned the hyper-parameters of each algorithm separately. Are the tuning attempts equal for each algorithm to have a fair comparison?

** Please note that the reviewer's current grade can be upgraded if the authors address the reviewer's main concerns written as above


**Time Spent Reviewing:**

7

---

> ### Author Response · Authors · 2021-08-08
> **Answering questions**
>
> Dear reviewer, thank you for your feedback which we believe is valuable and will help us to improve our submission. Please find detailed answers to your questions below. We would be grateful to hear whether this addresses your concerns, thank you.
>
> 2- SOSP:
> Yes, Assumption 1 covers the positive semidefinite case. Please compare Equation (1) in our submission to the definition given on page 2 of Jin et al. (https://arxiv.org/pdf/1703.00887.pdf). These are identical. As mentioned in Jin et al., under the assumption that all saddle points are strict then all second-order stationary points are local minima and our analysis then implies a stronger result. We will add a note in the text.
>
> 3- The contributions page 2:
> We will rewrite the list of contributions as suggested, thank you
>
> 4- Line 67:
> Yes, this is a typo, thank you for pointing this out!
>
> 5- Algorithm 1:
> Please check the beginning of section 3.1 where we explained why the two steps are required. The directions of the steps are sampled in the same way but the step size is different. Essentially, Algorithm 1 is a variant of STP with adaptive step sizes. Therefore, the two-step version we analyze in the paper is strictly better than STP, which implies that the exponential dependency to the dimension applies to STP.
>
> 6- Lemma 3:
> Please note that the definition of strict saddle does not rule out the presence of one or several zero eigenvalues, it just says $\lambda_{min}(\nabla^2 f(x)) < 0$, i.e. the minimum eigenvalue is negative but some eigenvalues can still be zero. Please see Jin et al. or https://bair.berkeley.edu/blog/2017/08/31/saddle-efficiency/. The intuition is as follows: as long as there is a direction of negative curvature, the algorithm is able to exploit this direction to decrease the objective function. The other directions in the space are not used in the worst-case analysis. We hope this clear out your confusion and we will make sure to add a note in the text.
>
> 7- Algorithm 2:
> Please note that the caption of Algorithm 2 states “Every iteration of DFPI requires 4d function evals”. The reason we wanted to highlight this fact is that this is a linear dependency to the dimension $d$. In contrast, vanilla STP suffers from an **exponential dependency**, which is a lot worse. Therefore, the cost of Algorithm 2 is a lot cheaper than STP and other similar approaches. Note that our main result (Thm 10) is a statement on the number of function evaluations, and hence takes into account this factor.
>
> 8- Assumption 2:
> This is an important point, note that we can guarantee that this assumption is satisfied. Please see Lemma 9 where we give a precise characterization of the number of iterations of DFPI required to ensure the validity of Assumption 2.
> Furthermore, we invite the reviewer to check Figures 10-12 in the appendix that provides additional empirical evidence about the validity of this assumption being satisfied.
>
> 9-  numerical experiments:
> We designed our experiments in order to verify the validity of the theoretical bounds derived in the paper. We have two sets of experiments:
> 1) First set of experiments is on relatively simple functions where the hyper-parameter tuning is relatively straightforward (unlike more complex problems discussed below). We note that our experimental setting follows common practices in the literature, see e.g. Flokas et al. Neurips 2019, https://arxiv.org/pdf/1910.13021.pdf (only a single toy problem). This also goes beyond the experimental settings of many derivative-free papers published at Neurips, e.g.:
> https://papers.nips.cc/paper/2020/file/6646b06b90bd13dabc11ddba01270d23-Paper.pdf
> https://papers.nips.cc/paper/2018/file/36d7534290610d9b7e9abed244dd2f28-Paper.pdf
> https://papers.nips.cc/paper/2012/file/e6d8545daa42d5ced125a4bf747b3688-Paper.pdf
>
> Therefore, we believe our experimental setting is in line with the current literature.
>
> 2) Second set of experiments is on a leading eigenvector problem, which is a fundamental primitive of numerical linear algebra with numerous applications. In machine learning, such applications include spectral clustering, community detection, etc. We point out that this problem is of great interest in optimization (https://proceedings.neurips.cc/paper/2018/file/1b318124e37af6d74a03501474f44ea1-Paper.pdf ), since the corresponding landscape is non-convex (see e.g. http://proceedings.mlr.press/v130/alimisis21a/alimisis21a.pdf )
>
> 10- Figure 3:
> The setup procedure is clarified in the “Setup” paragraph in Section 4. In each set of experiments, all algorithms are initialized at the same strict saddle point. Wherever possible, a different strict saddle point is used as initialization between different experimental runs (i.e. when repeating the experiments in order to compute confidence intervals).
>
> In Figure 3, AHDS and BDS do not reduce the objective function since they do not escape the saddle point. This can also be confirmed by inspecting the plot of the gradient norm in Figures 14 and 15 in the Appendix.
>
>
> Note that BDS does not have any theoretical convergence guarantee to SOSPs. Even if it evades saddle points, the speed at which it escapes could be very slow.
>
> We have two conjectures as to why AHDS does not escape the saddle in some experiments. 1) We did not find the optimal set of parameters in our grid search. 2) AHDS uses the same step size η_κ for both: i) Approximating Hessians (Step 6, Algorithm 8) and ii) Descending the loss (Step 1, Algorithm 8). The unique step makes AHDS less flexible. Instead, RSPI decouples this and uses independent parameters for descending the loss and approximating curvature.  We will try to investigate this further.
>
> Appendix D.2 (Figures 16 and 17) corresponds to a different set of experiments (Rastrigin function) compared to the results in Figure 3 (Function with growing dimension). For the results in Appendix D.2 (Rastrigin function), the initialization procedure is described in paragraph “Rastrigin function” in Section 4 and in greater detail in Appendix D.2. For this task, Appendix D.2 explains in detail how the initial point x_0 is selected as well as the impact that initialization has on the results of this task.
>
> 11- Tuned of the hyper-parameters:
> The selection of hyperparameters is first briefly described in the “Setup” paragraph in Section 4 and in more detail in Appendix F. In each experiment, the hyperparameters of every algorithm are selected based on a coarse grid-search that is refined heuristically by trial and error. The main hyperparameters that are considered for tuning are σ1, σ2 for RS and RSPI, η_0 and the update rule φ(.) for STP, η_0, η_max, γ and θ for BDS and AHDS (see Algorithms 6, 7 and 8 for the parameters of STP, BDS and AHDS respectively). For these parameters, a coarse grid-search was used to tune their value. We select the parameter values that give rise to the smallest objective value. In some cases, we also inspect the evolution of the optimality gap as a function of iterations while tuning hyperparameters. Whenever necessary, the grid-search is refined heuristically based on the outcome at hand and the trial procedure is repeated. Appendix F includes more details regarding the scheme that is used to update the parameters σ1 and σ2 of RS and RSPI as well as an illustration of the effect that certain crucial hyperparameters have on the performance of these two methods (see Figures 20 and 21).
>
> We try to give as much detail as possible in the appendix regarding the grid-search used to set the hyperparameters. This is of course not a problem specific to our paper, every paper has to deal with such a process.
> Importantly, we note that in all experiments, the parameters of RSP and RSPI are initialized and updated in the same manner. Hence, the only difference between the two methods is that RSPI explicitly extracts negative curvature whereas the two-step RS samples a direction at random.

---

> > ### Author Response · Authors · 2021-08-23
> > **Update**
> >
> > Dear reviewer,
> >
> > Since you mentioned that you will reconsider your score if we address your main concerns, we would be grateful to get some feedback on the answers we provided. We are happy to clarify any point if needed.
> >
> > Thank you very much,
> > The authors

---

> > > ### Comment · Reviewer_Tejo · 2021-09-01
> > > **Upgrading my score**
> > >
> > > Thanks the reviewer for the comprehensive answers to the reviewer's concerns. As the authors mentioned, they should apply the above comments in their manuscript. Particularly, it is highly suggested that the authors describe the steps of Algorithm 1 more clearly in the text. Overall, the authors addressed the reviewer's main concerns and the reviewer is eager to upgrade his score.

---

### Comment · Area_Chair_5nfk · 2021-09-18
**Comparison to previous work**

Dear Authors,

I apologize for the late response here, but the discussion brought up the following paper which I see you have cited -

https://arxiv.org/pdf/1910.13021.pdf - Efficiently avoiding saddle points with zero order methods: No gradients required

However I believe the comparison you have provided with this paper is very short. Could you clarify what exactly is novelty of your paper compared to this work?

Best
AC

---

> ### Author Response · Authors · 2021-09-19
> **Flokas et al.**
>
> Dear AC,
>
> Thank you for the opportunity to clarify this point.
>
> As we briefly discussed in the paper, the work of Floukas et al. belongs to the category of approaches that approximate the gradient using finite difference, see equation (1) in the arxiv version. In contrast, our work analyzes the convergence properties of random search methods, i.e. methods that sample a random direction. As you can see in Algorithm 1, two directions $s_1$ and $s_2$ are sampled uniformly from the sphere and then used to improve the function value. These are therefore different algorithms. Below, we explain why this difference is important and why our analysis is fundamentally different from Flokas et al.
>
> ## Why is this distinction important?
> The analysis of Flokas et al. shows that gradient-free methods that approximate the gradient match the convergence rate guarantees of their exact gradient-based counterparts (up to constants). However, existing lower bounds clearly show that random search methods have a worst rate of convergence (see paragraph “Lower bounds” in the related work section). Of special interest to us is the complexity w.r.t. the dimension of the problem, which is **exponential** for random search methods.
>
> ## Motivation for analyzing random search methods
> As we motivate in the introduction, direct search methods have played an important role in the field of optimization, see for instance reference [37] that has over 850 citations on Google scholar. Random search methods have recently become popular in reinforcement learning (RL), see e.g. references [39] or [40] (the latter has 210 citations). The possible reason as to why these methods are currently becoming more popular in RL is probably due to their ability for exploration. Unlike gradient-based methods, they have the ability to potentially finding different descent directions that could avoid various bad local minima. However, as we explained above, this comes at a **high price due to the curse of dimensionality**.  Our motivation is therefore to address this high complexity problem in order to make these methods more scalable. We also note that these methods also have additional benefits since they are easy to run in parallel, although this is not something we explore in our experiments due to limited computational resources (but this is certainly something bit IT companies have explored in the papers we cited above).
>
> ## How is the analysis different?
> Because Flokas et al. rely on an approximation of the gradient, they can follow a similar strategy employed to show that gradient-based methods escape saddle points (e.g. the work by Jin et al.). More precisely, they define the error between the approximate gradient $q_x(x_k, h_k)$ and the exact gradient $\nabla f(x_k)$ as:
> $\epsilon_k = q_x(x_k, h_k) - \nabla f(x_k).$
> They then require this error to be bounded, see page 7:
> $$
> \| q(x, h) − \nabla f(x) \| ≤ c_h|h|,
> $$
> where $c_h$ is a constant and $h$ is the size of the step used in the finite difference formula. The constant $c_h$ is directly controlled by $h$ and can be made as small as possible to better approximate the result of the gradient-based method.
>
> In our case, the update direction is sampled at random and we therefore do not have access to a parameter that allows us to better approximate the gradient “on demand”. Instead, we have to rely on a different proof technique that involves more **probabilistic geometric arguments in high-dimensional spaces** (see for instance Appendix A.1).
>
> We hope this sufficiently clarifies the difference with the work of Flokas et al. If accepted, we would of course extend the explanation explaining in detail how our approach differs from their work. We are also happy to provide further explanations if needed.
>
> Thank you,
> The authors

---

### Decision · Program_Chairs · 2021-09-27

**Decision:**

Accept (Poster)

**Comment:**

The paper considers the problem of finding second order stationary points via only functional evaluations. The paper considers specifically random search based methods and demonstrates a random search method that establishes convergence to SOSPs within O(d/eps^2) function evaluations. The analysis and results are solid and the algorithmic contribution is strong. The main criticism towards the paper is that of novelty of results compared to existing results on zeroth order optimization convergence to SOSPs which have been shown to achieve the same rate. The relative novelty here is that the paper focuses specifically on random search method as opposed to approximating gradient type of methods that exist in literature. There are benefits of random search methods as elucidated by the authors.

Overall the paper is right on borderline. I am recommending accept based on the reviewers' unanimous agreement with its contributions. I strongly suggest the authors to do a very clear comparison with Flokas et al stating their result and the comparison of their result with it.